# Prefoldin 5 is a microtubule-associated protein that suppresses Tau aggregation and neurotoxicity

Anjali Bisht[1], Srikanth Pippadpally[1], Snehasis Majumder[2], Athulya T Gopi[1], Abhijit Das[2], Chandan Sahi[1], Mani Ramaswami[3], Vimlesh Kumar[1]*

[1]Department of Biological Sciences, Indian Institute of Science Education and Research (IISER) Bhopal, Bhopal, India; [2]Department of Bioscience and Biotechnology, Indian Institute of Technology Kharagpur, Kharagpur, India; [3]Trinity College Institute of Neuroscience (TCIN), University of Dublin, Dublin, Ireland

## eLife Assessment

This work details the finding that in at least one of the subunits of the heterohexameric chaperone complex Pfdn5 has additional functions beyond its contribution to cytoskeletal protein folding in Drosophila. The authors provide **convincing** evidence that it is a hitherto unknown microtubule associated protein in addition to regulating microtubule organization and levels of tubulin monomers. The **important** findings show that Pfdn5 loss exaggerates pathological manifestations of mutant human Tau bearing FTDP-17 linked mutations in Drosophila, while its overexpression suppresses them, suggesting that the latter may constitute a future therapeutic approach.

**Abstract** Tauopathies represent a major class of neurodegenerative disorders associated with intracellular aggregates of the microtubule-associated protein Tau. To identify molecular modulators of Tau toxicity, we used a genetic screen to identify protein chaperones whose RNAi-mediated knockdown could modulate hTau$^{V337M}$-induced eye-ommatidial degeneration in *Drosophila*. This screen identified the Prefoldins Pfdn5 and Pfdn6 as strong modifiers of hTau$^{V337M}$ cytotoxicity. Consistent with the known function of Pfdn as a cotranslational chaperone for tubulin, *Pfdn5* mutants showed substantially reduced levels of tubulin monomer. However, additional microtubule-related functions were indicated by the robust unexpected association of Pfdn5 with axonal microtubules in vivo, as well as binding with stabilized microtubules in biochemical assays. Loss of Pfdn5 resulted in neuromuscular junctions (NMJ) defects similar to those previously described in hTau-expressing flies: namely, increased supernumerary boutons and fewer microtubule loops within mature presynaptic boutons. Significantly, synaptic phenotypes caused by hTau$^{V337M}$ overexpression were also strongly enhanced in a *Pfdn5* mutant background. Consistent with a role in modulating Tau toxicity, not only did loss of *Pfdn5* result in increased accumulations of Tau aggregates in hTau$^{V337M}$-expressing neurons, but also neuronal overexpression of Prefoldin strikingly ameliorated age-dependent neurodegeneration and memory deficits induced by pathological hTau. Together, these and other observations described herein: (a) provide new insight into Prefoldin-microtubule interactions; (b) point to essential post-translational roles for Pfdn5 in controlling Tau toxicity in vivo; and (c) demonstrate that Pfdn5 overexpression is sufficient to restrict Tau-induced neurodegeneration.

*For correspondence: vimlesh@iiserb.ac.in

Sent for Review 29 October 2024
Preprint posted 30 October 2024
Reviewed preprint posted 23 December 2024
Reviewed preprint revised 14 November 2025
Version of Record published 14 January 2026

## Introduction

Aberrant accumulation of misfolded protein aggregates is associated with neuroinflammation, neuronal death, and progressive cognitive decline in diverse groups of neurodegenerative diseases (*Ross and Poirier, 2004*; *Sweeney et al., 2017*; *Soto and Pritzkow, 2018*; *Lashuel, 2021*). Alzheimer's disease, as well as a subset of other neurodegenerative disorders, together referred to as Tauopathies, are defined by accumulated intracellular aggregates of the Tubulin-associated unit (Tau) protein (*Iqbal et al., 2010*; *Jouanne et al., 2017*; *Pinzi et al., 2023*). Despite substantial clinical interest and decades of research, effective therapeutic interventions for treating Tauopathies are still unavailable (*Khanna et al., 2016*; *Lashuel, 2021*).

Tau is predominantly expressed in neurons, where it stabilizes microtubules, thus facilitating intra-axonal transport (*Avila et al., 2004*; *Kent et al., 2020*; *Robbins et al., 2021*). Several mutations in the Tau protein have been identified that contribute to a wide spectrum of Tauopathies, including Alzheimer's disease, Pick's disease, progressive supranuclear palsy (PSP), corticobasal degeneration (CBD), and frontotemporal dementia with Parkinsonism linked to chromosome 17 (FTDP-17) (*Goedert and Jakes, 2005*; *Wolfe, 2009*). These pathogenic mutations enhance the propensity for Tau protein hyperphosphorylation at Ser/Thr residues, leading to the formation of neurofibrillary tangles via self-aggregation (*Tzioras et al., 2023*). The phosphorylated Tau disengages from the microtubule, potentially altering axonal transport and contributing to synapse loss and/or axon retraction (*Guo et al., 2020*; *Tzioras et al., 2023*). Thus, the self-aggregation of Tau and destabilization of microtubules may contribute to the progression of Tau pathogenesis. Such a model is supported by studies in *Drosophila* and rodent models of Tauopathies. Several of these models of tauopathy show disrupted microtubules, synaptic abnormalities, and abnormal motor behavior (*Stubbs et al., 2023*). Significantly, pharmacological stabilization of microtubules or reducing Tau levels can revert at least some of the defects observed in these Tauopathy models (*Zhang et al., 2005*; *Brunden et al., 2010*). However, alternative approaches to mitigate Tau-induced neurodegeneration are required because the currently available microtubule-targeting drugs are toxic at concentrations required to have an effect in the brain (*Yu et al., 2021*). One approach, suggested by several studies demonstrating a role for chaperone systems in Tauopathies (*Perez et al., 1991*; *Renkawek et al., 1994*; *Ostapchenko et al., 2013*), is to identify and manipulate specific molecular chaperones that directly or indirectly control Tau aggregation and Tau-induced neurotoxicity in vivo (*Blard et al., 2007*; *Darling et al., 2021*).

Molecular chaperones facilitate proper protein folding, prevent protein aggregation, and solubilize or facilitate autophagic or proteasomal elimination of protein aggregates (*Warrick et al., 1999*; *Dou et al., 2003*; *Buchner, 2019*). Consistent with this, enhanced expression of Hsp70 or HSP90 chaperones in mouse neuroblastoma N2A cells reduces pathological Tau levels by promoting the partitioning of Tau onto microtubules (*Dou et al., 2003*). On the other hand, as chaperones stabilize misfolded protein states, the expression of certain chaperones or cochaperones can sometimes also promote and facilitate the aggregation of Tau (*Bhattacharya et al., 2020*; *Criado-Marrero et al., 2021*). For instance, expression of HSP90 cochaperones, FKBP52, or Aha1 in the mouse brain enhances Tau aggregation, neuroinflammation, and cognitive decline in the Tau transgenic mouse model (*Shelton et al., 2017*; *Criado-Marrero et al., 2021*). These and other data indicate that: (a) chaperones not only alleviate but also aggravate Tau aggregation, and hence identification and analysis of chaperones that modulate Tau-aggregation and toxicity are required to understand biological and pathogenic mechanisms involved in Tauopathy, and (b) genetic or pharmacological manipulation of specific chaperone activities could be of possible therapeutic value for treating Tau-induced neurodegeneration.

Here, we report that Pfdn5 colocalizes with axonal microtubules and physically associates with stable microtubules. Loss of Pfdn5 resulted in a remarkable reduction in tubulin levels, disrupting microtubules in otherwise wild-type *Drosophila*, as well as the aggregation of Tau in axons and larval brain of the *Drosophila* hTau^V337M disease model. Moreover, Pfdn5 deletion exacerbates Tau-induced neurotoxicity, and overexpression of Pfdn5 mitigates the age-dependent progression of neurodegeneration and suppresses the learning and long-term memory deficits associated with Tau-induced neurotoxicity. These and other observations described in subsequent sections of this paper suggest that (1) In addition to its role as a cotranslational chaperone for tubulin, Pfdn5 has direct roles in the stability of mature microtubule filaments, and (2) Pfdn5 stabilizes microtubules, prevents neuronal loss, and delays the onset of Tau-induced neurotoxicity. Since the overexpression of Pfdn5 restored the

Tau-induced neurological abnormalities to the control levels without causing any detectable changes in synaptic morphology, cognitive impairment, or organismal health, we suggest that Pfdn5 could be a possible therapeutic target for Tauopathies.

## Results

### A reverse-genetic screen of *Drosophila* chaperones identified Prefoldins as genetic modifiers of Tau-induced neurodegeneration

To identify chaperones that modulate Tau-induced neurodegeneration, we performed a screen for chaperones whose RNAi-based knockdown would modify progressive cytotoxicity observed in the eyes of *Drosophila* expressing human Tau$^{V337M}$. We used the hTau$^{V337M}$ model because its expression in the eye resulted in moderate phenotypes and therefore, allowed us to score for both enhancement or suppression of eye-ommatidial degeneration visibly (*Chau et al., 2006*; *Blard et al., 2007*). We co-expressed this transgenic construct (*UAS-hTau$^{V337M}$*) with each of 109 RNAi lines (targeting 64 chaperones) in the *Drosophila* eye using the pan-retinal driver *GMR-Gal4* and examined how each RNAi line influenced ommatidial degeneration in hTau$^{V337\,M}$-expressing flies, seven days post-eclosion (*Figure 1—figure supplement 1*). We identified 20 chaperones that enhanced the neurotoxicity and 15 that suppressed hTau$^{V337M}$-induced ommatidial degeneration (*Figure 1*, *Figure 1—figure supplement 1*). Consistent with the previous studies (*Petrucelli et al., 2004*; *Kundel et al., 2018*), we found that the knockdown of Hsp70 enhanced Tau-induced neurodegeneration (*Figure 1—figure supplement 1I–I" and AH*), validating the authenticity of this screen. In addition, the screen identified novel candidate Tau modulators. These notably included *Drosophila* orthologs of Prefoldins, tubulin binding cofactor E (TBCE), and chaperonin containing TCP1 (CCT), chaperones known to co-translationally regulate proper folding of tubulin or actin monomers (*Figure 1C–L*). Knockdown of Prefoldin subunits by different independent RNAi constructs strongly enhanced the ommatidial degeneration (*Figure 1C–I* and Figure N). Similar effects were also seen following the knockdown of CCT or TBCE (*Figure 1J–L* and Figure N).

The effect of Pfdn5 knockdown on Tau-induced eye degeneration, measured by quantifying the percentage of the degenerated eye area, was particularly robust compared to the hTau$^{V337M}$-expressing flies (*GMR-Gal4 >UAS-hTau$^{V337M}$*: 42.74±1.59% vs *GMR-Gal4 >UAS-hTau$^{V337M}$*; *UAS-Pfdn5 RNAi*: 85.06±3.62%; *p*<0.001) (*Figure 1M*, *Figure 1—figure supplement 2*). Three independent Pfdn5 RNAi lines targeting different regions of Pfdn5 transcripts enhanced the Tau phenotypes (*Supplementary file 1*). The quantitative RT-PCR (qPCR) analysis revealed a reduction of ~50–60% *pfdn5* transcript upon knockdown with Actin5C-Gal4 (*Figure 1—figure supplement 2*). Moreover, prior cell culture studies have supported the idea that Prefoldin functions to regulate the solubility of aggregate-prone proteins (*Sakono et al., 2008*; *Sörgjerd et al., 2013*). For instance, the deletion of Pfdn is accompanied by the accumulation of PolyQ or Htt aggregates in cell lines (*Tashiro et al., 2013*; *Takano et al., 2014*). In order to explore whether this proposed role of Prefoldin could be relevant to the control of Tau aggregation and neurodegeneration in vivo, and the finding that Pfdn5 is downregulated in Alzheimer's patients (*Ji et al., 2022*; *Askenazi et al., 2023*), we selected Pfdn5 for careful and detailed analysis.

### Prefoldin 5 regulates microtubule organization and levels of tubulin monomers

Pfdn5 is a component of the hetero-hexameric Prefoldin complex, which regulates the folding of nascent actin and tubulin monomers (*Vainberg et al., 1998*; *Leroux et al., 1999*). To rigorously analyse the neuronal functions of Pfdn5, we generated loss-of-function mutants of Pfdn5 using CRISPR/Cas9-based genome editing. We created two independent *Pfdn5* mutants (*ΔPfdn5$^{15}$* and *ΔPfdn5$^{40}$*) using two distinct pairs of gRNAs (*Figure 2A*). Both these *Pfdn5* mutants were null alleles, as no *Pfdn5* transcript was detected in the mutants (*Figure 2—figure supplement 1*). Both homozygous and trans-heteroallelic mutants of Pfdn5 exhibited larval lethality at the L3 developmental stage (*Figure 2A*). These findings suggest that *Pfdn5* is an essential gene required ubiquitously for survival.

Analysis of third instar larval NMJ in *Pfdn5* mutants revealed the presence of several supernumerary boutons (*Figure 2—figure supplement 1B–I*). Prior work has shown that induction of supernumerary boutons can result from destabilizing the microtubule cytoskeleton at the NMJ (*Xiong*

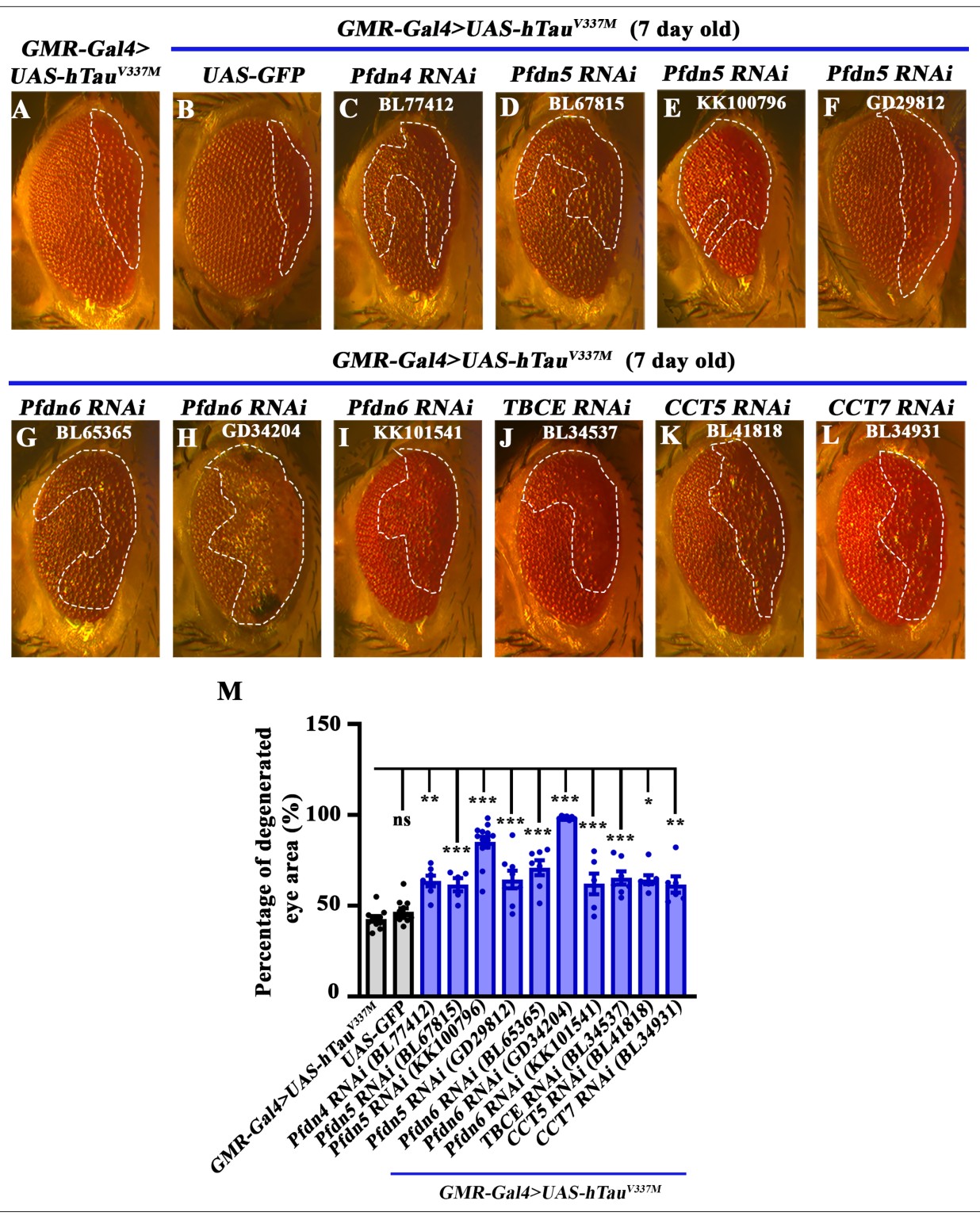

**Figure 1.** Prefoldin cochaperones are genetic modifiers of Tubulin-associated unit (Tau)-induced eye degeneration. (**A–L**) Bright-field images of 7-day-old *Drosophila* eyes expressing (**A**) *hTau^V337M* (control), (**B**) *GMR-Gal4>UAS-hTau^V337M; UAS-GFP* (Gal4 dilution control), (**C**) *GMR-Gal4>UAS-hTau^V337M; UAS-Pfdn4 RNAi*, (**D**), (**E**), (**F**) *GMR-Gal4>UAS-hTau^V337M; UAS-Pfdn5 RNAi*, (**G**), (**H**), (**I**) *GMR-Gal4>UAS-hTau^V337M; UAS-Pfdn6 RNAi*, (**J**) *GMR-Gal4>UAS-hTau^V337M; UAS-TBCE RNAi*, (**K**) *GMR-Gal4>UAS-hTau^V337M; UAS-CCT5 RNAi*, (**L**) *GMR-Gal4>UAS-hTau^V337M; UAS-CCT7 RNAi*. (**N**) Histogram showing the percentage of degenerated area in eyes of 7-day-old flies of genotypes: *UAS-hTau^V337M/+; GMR-Gal4/+* (42.74±1.59), *UAS-hTau^V337M/+; GMR-Gal4/UAS-GFP* (46.7±1.79), *UAS-hTau^V337M/+; GMR-Gal4/UAS-Pfdn4 RNAi* (BL77412; 63.9±2.95), *UAS-hTau^V337M/+; GMR-Gal4/UAS-Pfdn5 RNAi* (BL67815; 61.6±3.6), *UAS-hTau^V337M/+; GMR-Gal4/UAS-Pfdn5 RNAi* (KK100796; 85.06±3.01), *UAS-hTau^V337M/+; GMR-Gal4/UAS-Pfdn5 RNAi* (GD29812; 64.44±4.91),

*Figure 1 continued on next page*

*Figure 1 continued*

*UAS-hTau^V337M/+; GMR-Gal4/UAS-Pfdn6 RNAi (BL65365; 70.95±4.17), UAS-hTau^V337M/+; GMR-Gal4/+; UAS-Pfdn6 RNAi/+ (GD34204; 98.57±0.4), UAS-hTau^V337M/+; GMR-Gal4/UAS-Pfdn RNAi (KK101541; 77.11±2.96), UAS-hTau^V337M/+; GMR-Gal4/+; UAS-TBCE RNAi/+ (BL34537; 62.01±5.65), UAS-hTau^V337M/+; GMR-Gal4/+; UAS-CCT5 RNAi/+ (BL41818; 64.36±2.43), UAS-hTau^V337M/+; GMR-Gal4/+; UAS-CCT7 RNAi/+ (BL34931; 61.67±4.47). \*p<0.05; \*\*\*p<0.001; ns, not significant. At least 6 brightfield eye images of each genotype were used for quantification.*

The online version of this article includes the following source data and figure supplement(s) for figure 1:

**Source data 1.** Source data related to *Figure 1*.

**Figure supplement 1.** Screening of *Drosophila* chaperones to identify modifiers of Tauopathy.

**Figure supplement 1—source data 1.** Source data related to *Figure 1—figure supplement 1*.

**Figure supplement 2.** qPCR analysis for knockdown efficiency of RNAi lines against cytoskeleton regulatory chaperones.

**Figure supplement 2—source data 1.** Source data related to *Figure 1—figure supplement 2*.

*et al., 2013*; *Saunders et al., 2022*). We, therefore, investigated whether loss of Pfdn5 can influence axonal microtubules. We visualized microtubules using the monoclonal antibody 22C10, which labels the microtubule-associated protein Futsch in neurons. Futsch-positive loops are seen within a subset of stable presynaptic boutons (*Roos et al., 2000*). Boutons containing such loops were greatly reduced in the *Pfdn5* mutant (control: 19.98±2.18 vs *ΔPfdn5^15/40*: 7.72±1.62; *p*<0.001), and this reduction was restored to the control levels (*ΔPfdn5^15/40*: 7.72±1.62 vs *Elav-Gal4/+; UAS-Pfdn5/+; ΔPfdn5^15/40*: 27.39±2.21; *p*<0.001) upon pan-neuronal expression of a *Pfdn5* transgene in *Pfdn5* mutant background (*Figure 2B–F*). Consistent with this, we found a significant reduction in the intensity of acetylated tubulin, which represents long-lived, stable microtubules, at the synapses and in the muscle of *Pfdn5* mutant compared to the control (synapses - control: 0.54±0.04 vs. *ΔPfdn5^15/40*: 0.36±0.03: *p*<0.001; muscles - control: 530.1±71.56 vs. *ΔPfdn5^15/40*: 141.9±8.43: *p*<0.001) (*Figure 2—figure supplement 2A–H*). The reduced acetylated tubulin intensity at *Pfdn5* null mutant synapses was restored by the pan-neuronal expression of the *Pfdn5* transgene (*Figure 2—figure supplement 2A–H*). Taken together, the data support a function of Pfdn5 in regulating microtubule stability and organization in vivo.

Pfdn5 is also a known cotranslational chaperone for monomeric tubulin (*Zhang et al., 2016*; *Gestaut et al., 2022*). We, therefore, also examined whether and how loss of Pfdn5 altered levels of tubulin monomers. Using RT-PCR, we found that the transcript level of tubulin was not altered in the *Pfdn5* mutants, suggesting that Pfdn5 does not alter tubulin gene transcription or transcript stability (*Figure 2—figure supplement 2I*). In contrast, western blotting revealed about 85% reduction in the α-tubulin monomers (control: 1.00±0.00 vs *ΔPfdn5^15/40*: 0.11±04; *p*<0.001), about 70 % reduction in β-tubulin (control: 1.00±0.00 vs *ΔPfdn5^15/40*: 0.31±09 *p*<0.001) and about 60% reduction in ace-tubulin levels (control: 1.00±0.00 vs *ΔPfdn5^15/40*: 0.46±0.04; *p*<0.001) in the *Pfdn5* mutant compared to the control (*Figure 2G–J*). These reductions were not seen when a *Pfdn5* transgene was ubiquitously expressed in a *Pfdn5* mutant background (*Figure 2G–J*). The observation that β-actin levels remained unchanged (*Figure 2G*) in *Pfdn5* mutants is not particularly surprising, given a prior report indicating a differential requirement of the *Drosophila* Prefoldin complex in actin and tubulin biogenesis (*Delgehyr et al., 2012*). Together, these data are consistent with: (a) *Drosophila* Pfdn5 serving as an evolutionarily conserved function in cotranslational folding of monomeric tubulin; and (b) an additional role may be performed by Pfdn5 in the regulation of mature microtubule filaments in neurons. One or both of these functions could potentially be required for the stabilization of axonal microtubules.

## Pfdn5 is a novel neuronal microtubule-associated protein

To assess the function of Pfdn5, we began by generating antisera against the full-length Pfdn5. Western blot analysis using Pfdn5 antibody revealed a protein band of ~18 kDa in the larval lysates that was absent in *Pfdn5* mutants, thus validating the specificity of the antibody (*Figure 3—figure supplement 1A*). In larval fillets examined by immunocytochemistry, anti-Pfdn5 staining was dramatically reduced in the mutant, with faint residual staining consistent with a low level of maternally provided Pfdn5 (*Figure 3A–B"'* and *Figure 3—figure supplement 1B–E*). The Pfdn5 staining was restored upon expression of the Pfdn5 transgene in the Pfdn5 mutant background (*Figure 3B–C"'* and *Figure 3—figure supplement 1D–E*). However, heterozygous mutants of Pfdn5 (*ΔPfdn5^15/+* and *ΔPfdn5^40/+*) revealed no significant reduction in the levels of Pfdn5 or tubulin (*Figure 3—figure supplement*

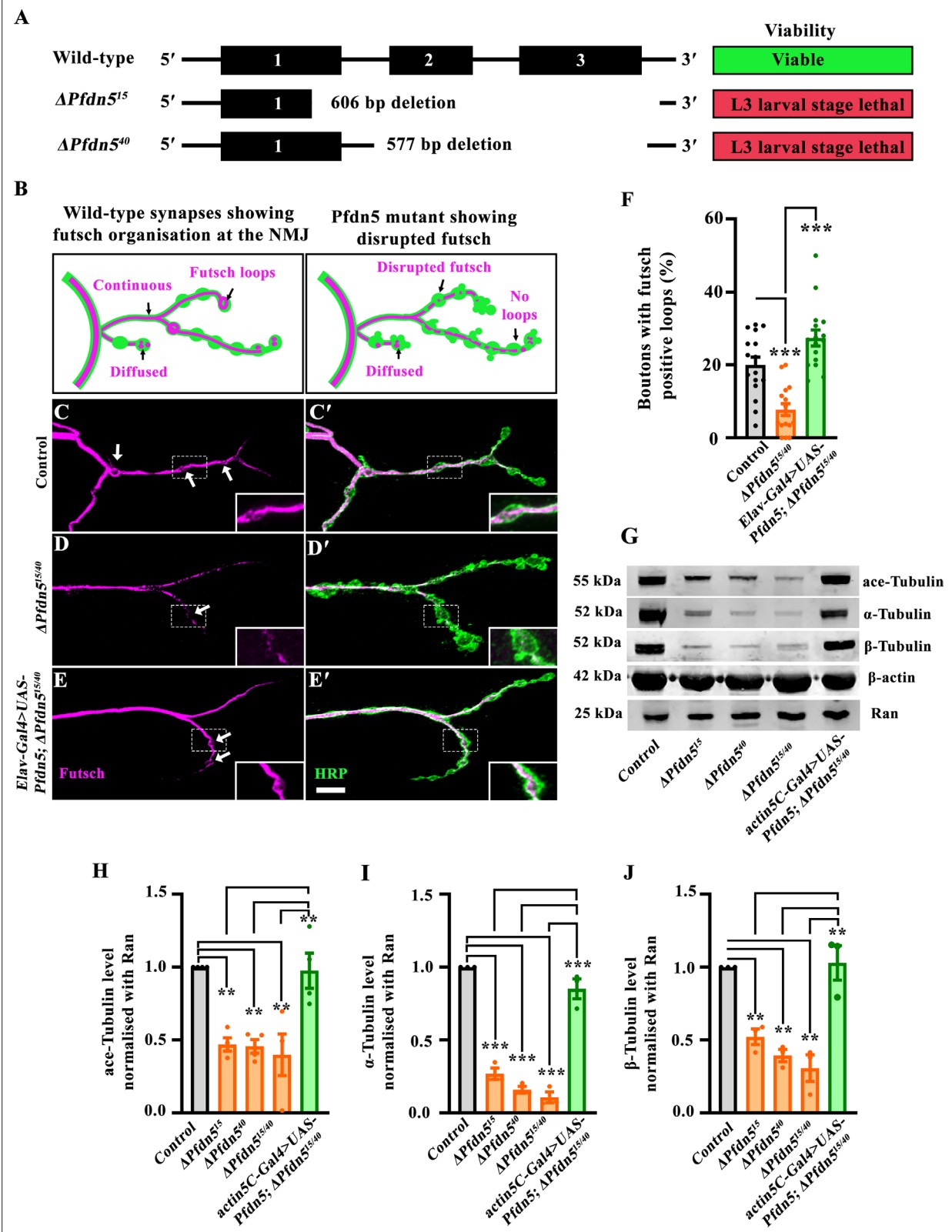

**Figure 2.** Loss of Pfdn5 disrupts microtubule organization. (**A**) Generation of a loss-of-function mutant of Pfdn5 using CRISPR/Cas9-based genome editing. Schematic representation of the Pfdn5 genomic organization showing exons (solid black boxes, 1–3) and introns (thin black lines). Two loss-of-function Pfdn5 mutants with 606 bp (line-15) or 577 bp (line-40) deletion were obtained. Both mutant lines are third-instar larval lethal. (**B**) Schematic representation of Futsch loop organization in muscle 4 of A2 hemisegment in wild-type or *Pfdn5* mutant. *Pfdn5* mutant shows diffused Futsch loop

*Figure 2 continued on next page*

*Figure 2 continued*

organization and reduced loops at the terminal boutons. (**C-E'**) Confocal images of NMJ synapses at muscle 4 of A2 hemisegment showing Futsch loops in (**C-C'**) control, (**D-D'**) *ΔPfdn5$^{15/40}$*, (**E-E'**) *Elav-Gal4>UAS-Pfdn5; ΔPfdn5$^{15/40}$* double immunolabeled with neuronal membrane marker, HRP (green) and 22C10 antibody against microtubule-associated protein, Futsch (magenta). The scale bar in E' for (**C-E'**) represents 10 µm. The inset shows a 2.0 x magnified Futsch loop. (**F**) Histogram showing the percentage of Futsch positive loops from muscle 4 at A2 hemi-segment in control (19.98±2.18), *ΔPfdn5$^{15/40}$* (7.72±1.62), *Elav-Gal4/+; UAS-Pfdn5/+; ΔPfdn5$^{15/40}$* (27.39±2.21). ***$p<0.001$. At least 16 neuromuscular junctions (NMJs) of each genotype were used for quantification. (**G**) Western blot showing protein levels of α-Tubulin, β-Tubulin, ace-Tubulin, and Actin in control, *ΔPfdn5$^{15/15}$*, *ΔPfdn5$^{40/40}$*, *ΔPfdn$^{15/40}$*, and *actin5C-Gal4>UAS-Pfdn5; ΔPfdn5$^{15/40}$*. Ran protein levels were used as an internal loading control. (**H**) Histogram showing the percentage of ace-Tubulin normalized with Ran in control (1.00±0.00), *ΔPfdn5$^{15/15}$* (0.50±0.05), *ΔPfdn$^{40/40}$* (0.49±0.04), *ΔPfdn5$^{15/40}$* (0.46±0.04), *actin5C-Gal4/UAS-Pfdn5; ΔPfdn5$^{15/40}$* (1.05±0.13). **$p<0.01$. Three independent western blots were used for quantification. (**I**) Histogram showing percentage of α-Tubulin normalized with Ran in control (1.00±0.00), *ΔPfdn5$^{15/15}$* (0.27±0.04), *ΔPfdn$^{40/40}$* (0.16±0.02), *ΔPfdn5$^{15/40}$* (0.11±0.04), *actin5C-Gal4/UAS-Pfdn5; ΔPfdn5$^{15/40}$* (0.85±0.07). ***$p<0.001$. Three independent western blots were used for quantification. (**J**) Histogram showing percentage of β-Tubulin normalized with Ran in control (1.00±0.00), *ΔPfdn5$^{15/15}$* (0.52±0.04), *ΔPfdn$^{40/40}$* (0.39±0.04), *ΔPfdn5$^{15/40}$* (0.30±0.09), *actin5C-Gal4/UAS-Pfdn5; ΔPfdn5$^{15/40}$* (1.03±0.11). **$p<0.01$. Three independent western blots were used for quantification.

The online version of this article includes the following source data and figure supplement(s) for figure 2:

**Source data 1.** Source data related to *Figure 2*.

**Source data 2.** PDF files containing original Western blots for *Figure 2G*, indicating the relevant bands.

**Source data 3.** Original files for Western blot analysis displayed in *Figure 2G*.

**Figure supplement 1.** *Pfdn5* mutants are null alleles with no detectable *Pfdn5* transcripts.

**Figure supplement 1—source data 1.** Source data related to *Figure 2—figure supplement 1*.

**Figure supplement 2.** Loss of Pfdn5 disrupts microtubule cytoskeleton.

**Figure supplement 2—source data 1.** Source data related to *Figure 2—figure supplement 2*.

*1F–H*). Careful examination shows that Pfdn5 colocalized with axonal microtubule (labeled with α-tubulin antibody) in wild-type larvae (*Figure 3A–A"'*). The level of α-tubulin was significantly reduced in the absence of Pfdn5 (*Figure 3A–C"'*). The Pearson's correlation coefficient of 0.60±0.02 across pixels labeled by α-tubulin and Pfdn5 in axons further strengthens that a tight colocalization exists between Pfdn5 and neuronal microtubules (*Figure 3D*).

To further test whether Pfdn5 associates with microtubules, we stabilized microtubules using Taxol and performed a microtubule-binding experiment (*Ando et al., 2016*). We found Pfdn5 in the pellet fraction when microtubules were stabilized with Taxol, but not under the condition where the microtubules were severed using Nocodazole. The quantification revealed a substantially higher Pfdn5 binding to stabilized microtubules when compared to non-stabilized microtubule control (Taxol: 36.67±7.56, vs control: 7.83±2.92; $p<0.001$) (*Figure 3E–G*). Consistent with our immunocytochemistry results, we found that Pfdn5 binds with the Taxol-stabilized microtubule. This unexpected localization of Pfdn5 to neuronal microtubule filaments and its binding to the stable microtubule points to a role for this protein beyond its function as a cotranslational chaperone, potentially in the organization or the stability of axonal microtubule cytoskeleton.

## Loss of Pfdn5 phenocopies and synergistically aggravates the Tau-induced synaptic defects

Since microtubules regulate the morphological features of synapses (*Roos et al., 2000*) and loss of Pfdn5 resulted in reduced stable microtubules, we next asked if *Pfdn5* mutants show distinctly altered synaptic architecture at their NMJ. Both the homozygous and heteroallelic mutant combination showed numerous supernumerary boutons with altered synaptic morphology when compared to the control (control: 2.25±0.41 vs. *ΔPfdn5$^{15/40}$*: 18.25±1.27; $p<0.001$) (*Figure 4A–B , and J*). Increased supernumerary boutons in the *Pfdn5* mutant were completely restored upon pan-neuronal expression of a wild-type *Pfdn5* transgene using *Elav-Gal4* in the *Pfdn5* mutant background (*ΔPfdn5$^{15/40}$*: 18.25±1.27 vs *Elav-Gal4/+; UAS-Pfdn5/+; ΔPfdn5$^{15/40}$*: 2.94±0.67; $p<0.001$) (*Figure 4C and J*), further confirming that these phenotypes were caused by *Pfdn5* mutations and not unknown potential background mutations. Expression of *Pfdn5* in muscles using *mef2-Gal4* failed to rescue the lethality or the synaptic defects (*ΔPfdn5$^{15/40}$*: 18.25±1.27 vs *UAS-Pfdn5/+; mef2-Gal4, ΔPfdn5$^{15/40}$*: 15.60±0.86; $p>0.75$) (*Figure 4D and J*), suggesting that the synaptic phenotype arises due to loss of Pfdn5 in neurons and not muscles.

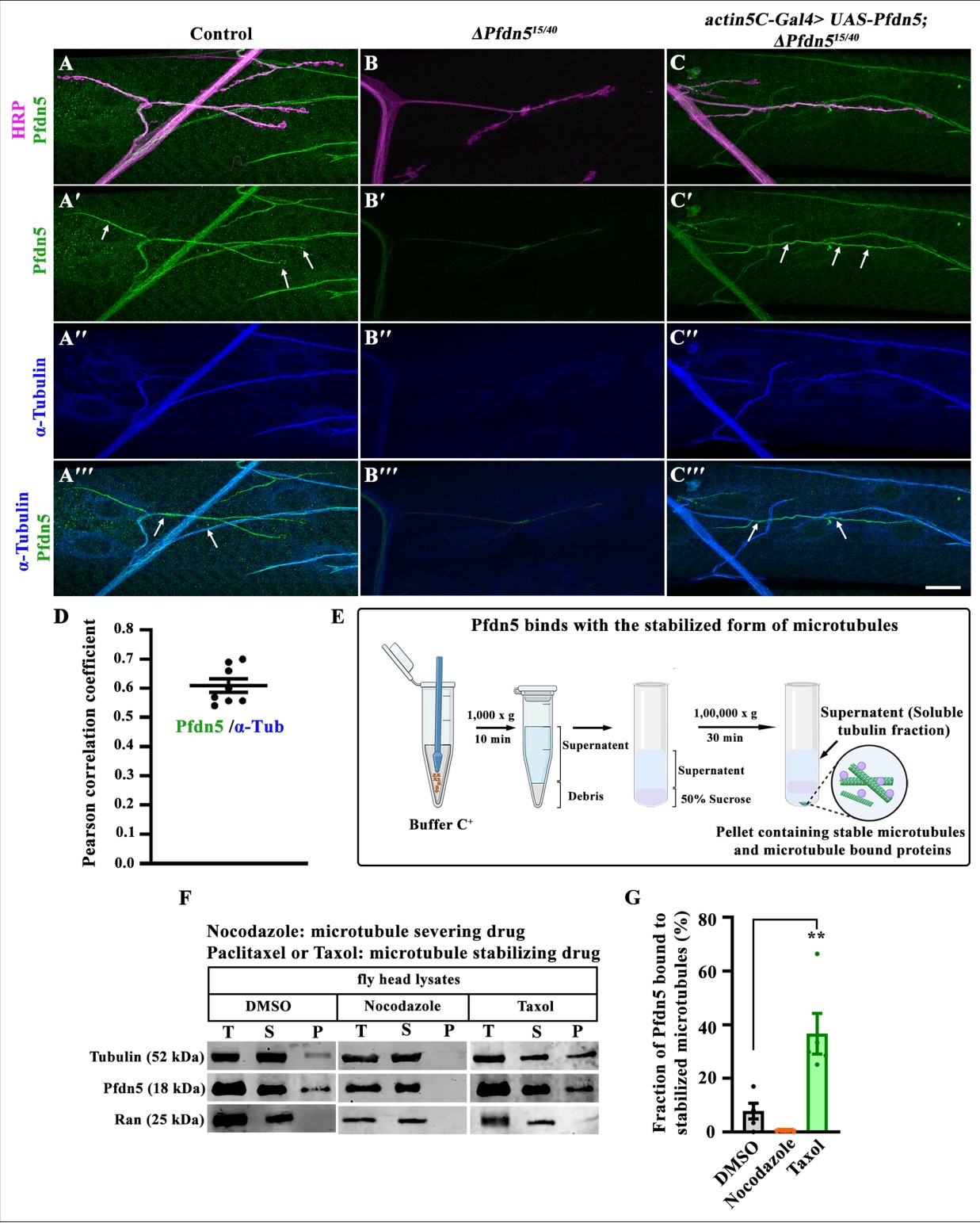

**Figure 3.** Pfdn5 is a novel microtubule-binding protein. (**A–C'''**) Confocal images of NMJ synapses at muscle 4 of A2 hemisegment triple-labeled for Pfdn5 (green), α-Tubulin (blue), and HRP (magenta) showing that Pfdn5 colocalizes with microtubule cytoskeleton in (**A-A'''**) in the wild-type larval neuronal axons and in the tracheal tubes, (**B-B'''**) *Pfdn5* mutants show dramatically reduced Pfdn5 and α-Tubulin levels, (**C-C'''**) *actin5C-Gal4*-mediated rescue (*actin5C-Gal4>UAS-Pfdn5; ΔPfdn5^{15/40}*) significantly restored the level of Pfdn5 and α-Tubulin. Arrows represent Pfdn5 colocalization with microtubule loops, which are not detectable in the *Pfdn5* mutants. The scale bar in C''' (for A-C''') represents 10 μm. (**D**) Pearson's correlation coefficient to quantify colocalization between Pfdn5 and axonal microtubule. (**E**) Schematic representation of the microtubule-binding protocol. Head lysate from

*Figure 3 continued on next page*

*Figure 3 continued*

wild-type flies, treated with Taxol or Nocodazole, was subjected to ultracentrifugation. The 'soluble fraction' contains free tubulin, whereas the 'insoluble pellet fraction' contains the stabilized microtubule along with microtubule-binding proteins, which can be detected by western blotting. The details of the protocol is described in the material and methods section. (F) Microtubule binding assay with *Drosophila* head lysate in the presence of Taxol or Nocodazole (diluted in DMSO). T represents (Total fraction: input fraction), S represents (Supernatant: free tubulin), and P represents (Pellet fraction: stabilized microtubule). Immunoblot with antibodies against ace-Tubulin or Pfdn5 detected increased Pfdn5 in the pellet fraction in the presence of Taxol but not Nocodazole. The binding of Pfdn5 with stabilized microtubules was calculated as the percentage of Pfdn5 in the pellet fraction. Ran was used as the loading control. (G) Histogram showing the percentage of the Pfdn5 in the pellet fraction of in vivo microtubule binding assay in the presence of DMSO (7.83±2.92), Nocodazole (0.42±0.1), or Taxol (36.67±7.56). Five independent western blots were used for quantification.

The online version of this article includes the following source data and figure supplement(s) for figure 3:

**Source data 1.** Source data related to *Figure 3*.

**Source data 2.** PDF files containing original Western blots for *Figure 3F*, indicating the relevant bands.

**Source data 3.** Original files for Western blot analysis displayed in *Figure 3F*.

**Figure supplement 1.** Generation and characterization of Pfdn5 antibody.

**Figure supplement 1—source data 1.** Source data related to *Figure 3—figure supplement 1*.

**Figure supplement 1—source data 2.** PDF files containing original Western blots for *Figure 3—figure supplement 1A and F*, indicating the relevant bands.

**Figure supplement 1—source data 3.** Original files for Western blot analysis displayed in *Figure 3—figure supplement 1A and F*.

Ectopic satellite boutons and disrupted microtubules that we observed in *Pfdn5* mutant larval NMJs appeared very similar to those previously described in *Drosophila* expressing hTau$^{V337M}$ in motor neurons (*Blard et al., 2007*; *Xiong et al., 2013*; *Mao et al., 2017*). This apparent similarity, together with our identification of Pfdn5 as a genetic modifier of hTau$^{V337M}$-induced cytotoxicity, led us to more closely examine phenotypic similarities between *Pfdn5* mutant and hTau$^{V337M}$-expressing animals, as well as genetic interactions between *Pfdn5* and hTau$^{V337M}$. We first confirmed that the loss of Pfdn5 phenocopies the Tau$^{V337M}$-induced morphological defects at synapses (*Figure 4E–F*). We then tested the effects of loss and gain of Pfdn5 on Tau$^{V337M}$ phenotypes.

Morphological NMJ phenotypes induced by expressing hTau$^{V337M}$ pan-neuronally using *Elav-Gal4* were strongly enhanced in a *Pfdn5* mutant background (*Figure 4G and I–K*). While satellite bouton numbers in *Elav-Gal4/UAS-hTau$^{V337M}$* and *ΔPfdn5$^{15/40}$* were 18.25±1.27 and 14.06±1.00, respectively, this was significantly increased in the *Elav-Gal4/UAS-hTau$^{V337M}$; ΔPfdn5$^{15/40}$* combination (32.25±3.22; $p<0.001$) (*Figure 4J*). Subsequently, total bouton number was significantly increased in *Elav-Gal4/UAS-hTau$^{V337M}$; ΔPfdn5$^{15/40}$* combination animals (60.24±3.76; $p<0.001$) compared to either *Elav-Gal4/UAS-hTau$^{V337M}$* (36.00±2.65) or *ΔPfdn5$^{15/40}$* (38.38±2.15) alone. Similarly, bouton area (in μm$^2$) at NMJs of *Elav-Gal4/UAS-hTau$^{V337M}$; ΔPfdn5$^{15/40}$* combination animals (2.29±0.20) were substantially smaller than either *Elav-Gal4/UAS-hTau$^{V337M}$* (6.40±0.45) or *ΔPfdn5$^{15/40}$* (5.09±0.28) alone (*Figure 4K*). Consistently, expressing a more severe form of pathological Tau (hTau$^{R406W}$) in the *Pfdn5* mutation background also resulted in further enhancement in the synaptic phenotypes (*Figure 4—figure supplement 1A–E*). Thus, loss of Pfdn5 aggravates not only Tau-induced eye degeneration (*Figure 1*) but also specific Tau-induced synaptic defects, suggestive of function in a common pathway.

Next, we investigated whether this enhanced synaptic phenotype results from increased microtubule disruption. We found that Futsch staining was significantly reduced at the synapses when hTau$^{V337M}$ was expressed in the *Pfdn5* mutant background (*Figure 4—figure supplement 1F–J*). These findings indicate that microtubule disruption caused by the loss of Pfdn5 contributes to the exacerbation of hTau$^{V337M}$-induced synaptic defects.

## Loss of Pfdn5 enhances pathological Tau aggregation in larval brain and axons

Tau-induced neurotoxicity directly correlates with the extent of Tau phosphorylation and its deposition as insoluble aggregates (*Chau et al., 2006*; *Aquino Nunez et al., 2022*). Hence, we performed additional experiments to explore mechanisms by which Pfdn5 levels could influence Tau function. We considered a model in which Pfdn5 acts to prevent Tau aggregation, thereby suppressing the Tau-induced neurodegeneration. The immunocytochemistry revealed that animals lacking Pfdn5 showed a remarkable increase in Tau-punctae structures in the larval brain (Tau punctae with size >3 μm$^2$ per

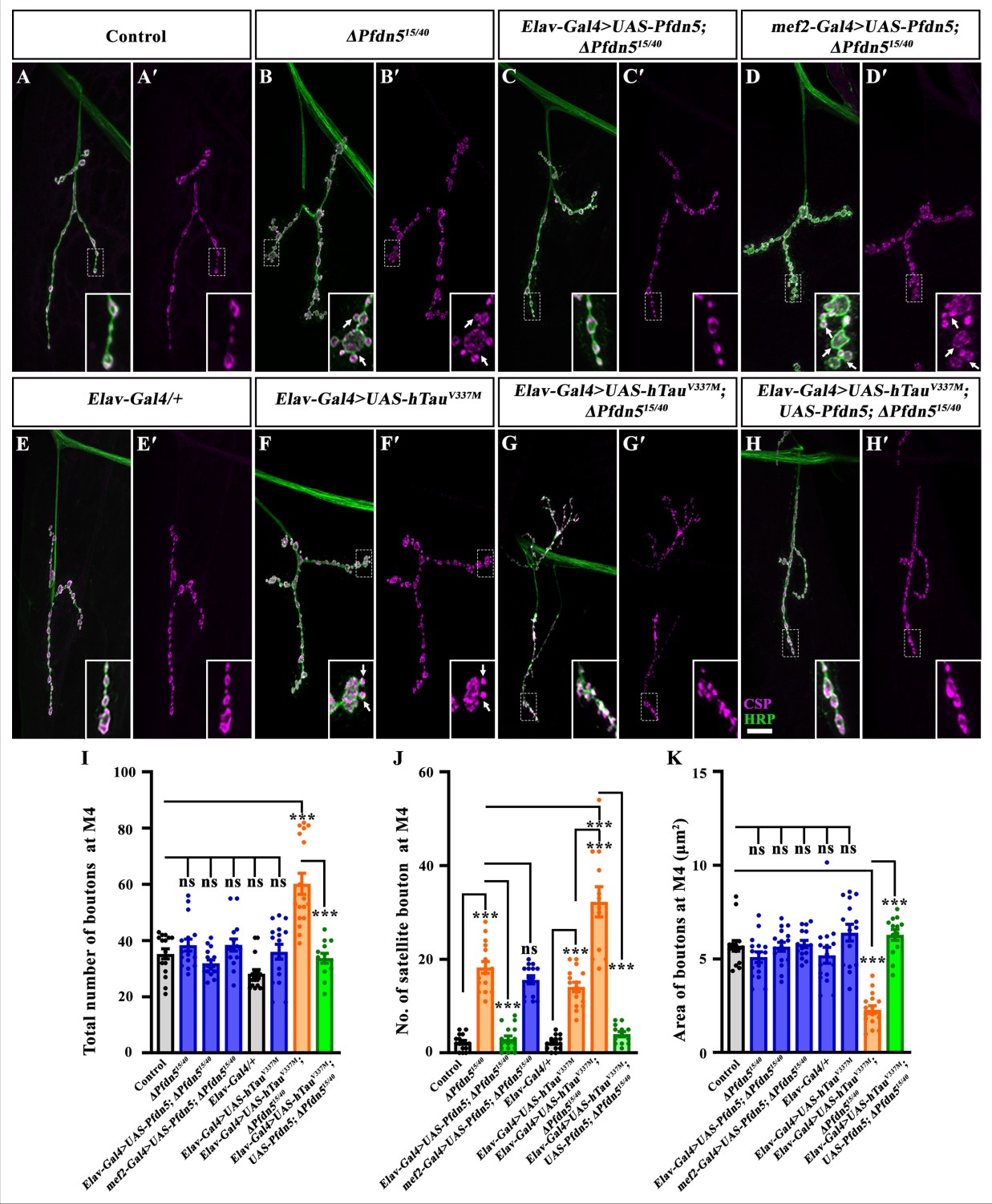

**Figure 4.** Loss of Pfdn5 mimics and enhances Tubulin-associated unit (Tau)-induced synaptic defects. (**A–H'**) Confocal images of neuromuscular junction (NMJ) synapses at muscle 4 of A2 hemisegment showing synaptic morphology in (**A-A'**) control, (**B-B'**) *ΔPfdn5*[15/40], (**C-C'**) *Elav-Gal4>UAS-Pfdn5; ΔPfdn5*[15/40], (**D-D'**) *mef2-Gal4>UAS-Pfdn5; ΔPfdn5*[15/40], (**E-E'**) *Elav-Gal4/+* (Gal4 control), (**F-F'**) *Elav-Gal4>UAS-hTau*[V337M], (**G-G'**) *Elav-Gal4>hTau*[V337M]; *ΔPfdn5*[15/40], (**H-H'**) *Elav-Gal4>hTau*[V337M];*UAS-Pfdn5; ΔPfdn5*[15/40] double immunolabeled for HRP (green), and CSP (magenta). The scale bar in H for (**A-H'**) represents 10 μm. Arrows point to clustered satellite boutons. (**I**) Histogram showing total number of boutons from muscle 4 at A2 hemisegment in control (35.25±1.8), *ΔPfdn5*[15/40] (38.38±2.15), *Elav-Gal4>UAS-Pfdn5; ΔPfdn5*[15/40] (31.94±1.18), *mef2-Gal4>UAS-Pfdn5; ΔPfdn5*[15/40] (38.40±2.17), *Elav-Gal4/+* (28.13±1.51), *Elav-Gal4>UAS-hTau*[V337M] (36.00±2.65), *Elav-Gal4>hTau*[V337M]; *ΔPfdn5*[15/40] (60.34±3.76), *Elav-Gal4>hTau*[V337M]; *UAS-Pfdn5; ΔPfdn5*[15/40]

*Figure 4 continued on next page*

*Figure 4 continued*

(33.69±1.76). ***p<0.001; ns, not significant. At least 12 NMJs of each genotype were used for quantification. (**J**) Histogram showing number of satellite boutons from muscle 4 at A2 hemisegment in control (2.25±0.41), *ΔPfdn5^{15/40}* (18.25±1.28), *Elav-Gal4>UAS-Pfdn5; ΔPfdn5^{15/40}* (2.94±0.67), *mef2-Gal4>UAS-Pfdn5; ΔPfdn5^{15/40}* (15.6±0.86), *Elav-Gal4/+* (2.2±0.38), *Elav-Gal4>UAS-hTau^{V337M}* (14.06±1.00), *Elav-Gal4>hTau^{V337M}; ΔPfdn5^{15/40}* (32.25±3.2), *Elav-Gal4>hTauV^{V337M};UAS-Pfdn5; ΔPfdn5^{15/40}* (4.0±0.5). ***p<0.001; ns, not significant. At least 12 NMJs of each genotype were used for quantification. (**K**) Histogram showing bouton area from muscle 4 at A2 hemi segment in control (5.7±0.29), *ΔPfdn5^{15/40}* (5.1±0.28), *Elav-Gal4>UAS-Pfdn5; ΔPfdn5^{15/40}* (5.6±0.2), *mef2-Gal4>UAS-Pfdn5; ΔPfdn5^{15/40}* (5.8±0.2), *Elav-Gal4/+* (5.2±0.4), *Elav-Gal4>UAS-hTau^{V337M}* (6.4±0.4), *Elav-Gal4>hTauV^{V337M}; ΔPfdn5^{15/40}* (2.3±0.2), *Elav-Gal4>hTauV^{V337M};UAS-Pfdn5; ΔPfdn5^{15/40}* (6.3±0.3). ***p<0.001; ns, not significant. At least 12 NMJ of each genotype were used for quantification.

The online version of this article includes the following source data and figure supplement(s) for figure 4:

**Source data 1.** Source data related to *Figure 4*.

**Figure supplement 1.** Loss of Pfdn5 enhances hTau^{R406W}-induced synaptic phenotypes.

**Figure supplement 1—source data 1.** Source data related to *Figure 4—figure supplement 1*.

brain lobe: *Elav-Gal4/UAS-hTau^{V337M}*: 1.13±0.39 vs *Elav-Gal4/UAS-hTau^{V337M}; ΔPfdn5^{15/40}*: 10.50±2.57; *p<0.001*) (*Figure 5A–F*). Consistent with these observations, hTau distribution in *Pfdn5* mutant axons revealed a substantially higher number of hTau-punctae compared to animals with normal levels of Pfdn5 (*Elav-Gal4/UAS-hTau^{V337M}*: 0.25±0.09/100 μm$^2$ vs *Elav-Gal4/UAS-hTau^{V337M}; ΔPfdn5^{15/40}*: 2.9±0.41/100 μm$^2$; *p<0.001*) (*Figures 4K and 5G-J''*). The increased hTau puncta in *Pfdn5* mutants were significantly suppressed upon normalizing the level of Pfdn5 in neurons. Analysis of fluorescence intensity profiles across the Tau puncta showed a fourfold increase in Tau intensity, further supporting that Tau indeed forms aggregates in the absence of Pfdn5 (*Figure 5—figure supplement 1A–C*). Additional experiments involving quantification of axonal hTau using immunofluorescence revealed that levels of Tau were significantly reduced in animals overexpressing hTau^{V337M} in *Pfdn5* mutants compared to animals expressing hTau^{V337M} alone (*Figure 5G–J'' and L*). Similar results were obtained when the phospho-Tau antibody was used to assess Tau levels and aggregates (*Figure 5—figure supplement 1D–F*). To determine if loss of Pfdn5 specifically modulates FTDP-17-associated Tau mutations or also influences sporadic Tauopathy linked to wild-type Tau, we expressed hTau^{WT} in *Pfdn5* mutants. Interestingly, expression of wild-type Tau (hTau^{WT}) in the *Pfdn5* mutants showed an aggregation pattern in the larval optic lobes similar to that observed with Tau^{V337M} (*Figure 5—figure supplement 1G–I*). These data reveal that Pfdn5 suppresses Tau aggregation in both the brain and axons and that loss of Pfdn5 can induce the onset of multiple forms of Tauopathies.

Next, in order to assess the nature of Tau aggregates, we performed a Tau solubility assay (*Vourkou et al., 2023*). We found that in the absence of Pfdn5, hTau levels were reduced in the supernatant fraction and increased in the pellet fraction (*Figure 5—figure supplement 2A–C*). Previous in vitro studies have shown that hTau can undergo liquid-liquid phase separation to form Tau droplets, which can be dissolved by 1,6-Hexanediol; in contrast, stable aggregates of Tau remain intact in the presence of 1,6-Hexanediol (*Wegmann et al., 2018*). To determine whether the Tau puncta formed in the absence of Pfdn5 represented stable aggregates or phase-separated condensates, we treated the larval fillets with increasing concentrations of 1,6-hexanediol. We observed that the Tau puncta formed in *Pfdn5* mutants remain intact even at 5% 1,6-Hexanediol (*Figure 5—figure supplement 2D–L*), indicating that loss of Pfdn5 promotes the formation of stable Tau aggregates.

Colocalization analysis with the HRP and Elav showed that these aggregates are mainly present in the cell body and axons (*Figure 5—figure supplement 2D–N'*). Tau aggregation is known to occur either through Tau hyperphosphorylation or disruption of microtubules (*Jackson et al., 2002*; *Okenve-Ramos et al., 2024*). To examine the role of phosphorylation, we stained the aggregates with the phospho-Tau-specific antibody, AT8, which detects pSer202 and pThr205 (*Goedert et al., 1995*). The AT8 antibody failed to label all aggregates detected by the total Tau antibody D5D8N, suggesting that Tau aggregates formed in the loss of Pfdn5 animals might have conformational heterogeneity (*Figure 5—figure supplement 3A–E*). Next, we inhibited Tau phosphorylation using 20 mM LiCl (*Cowan et al., 2010*) in *Elav-Gal4 >UAS-hTau^{V337M}; ΔPfdn5^{15/40}* larvae. We found that LiCl treatment did not significantly alter aggregate formation, indicating a minimal role of Tau hyperphosphorylation in the formation of aggregates in the absence of Pfdn5 (*Figure 5—figure supplement 3F–J*). Together, (a) the marked increase of aggregated hTau in the absence of Pfdn5 and (b) enhancement of the Tau-associated phenotypes by loss of Pfdn5 indicate that Pfdn5 prevents the transition of hTau

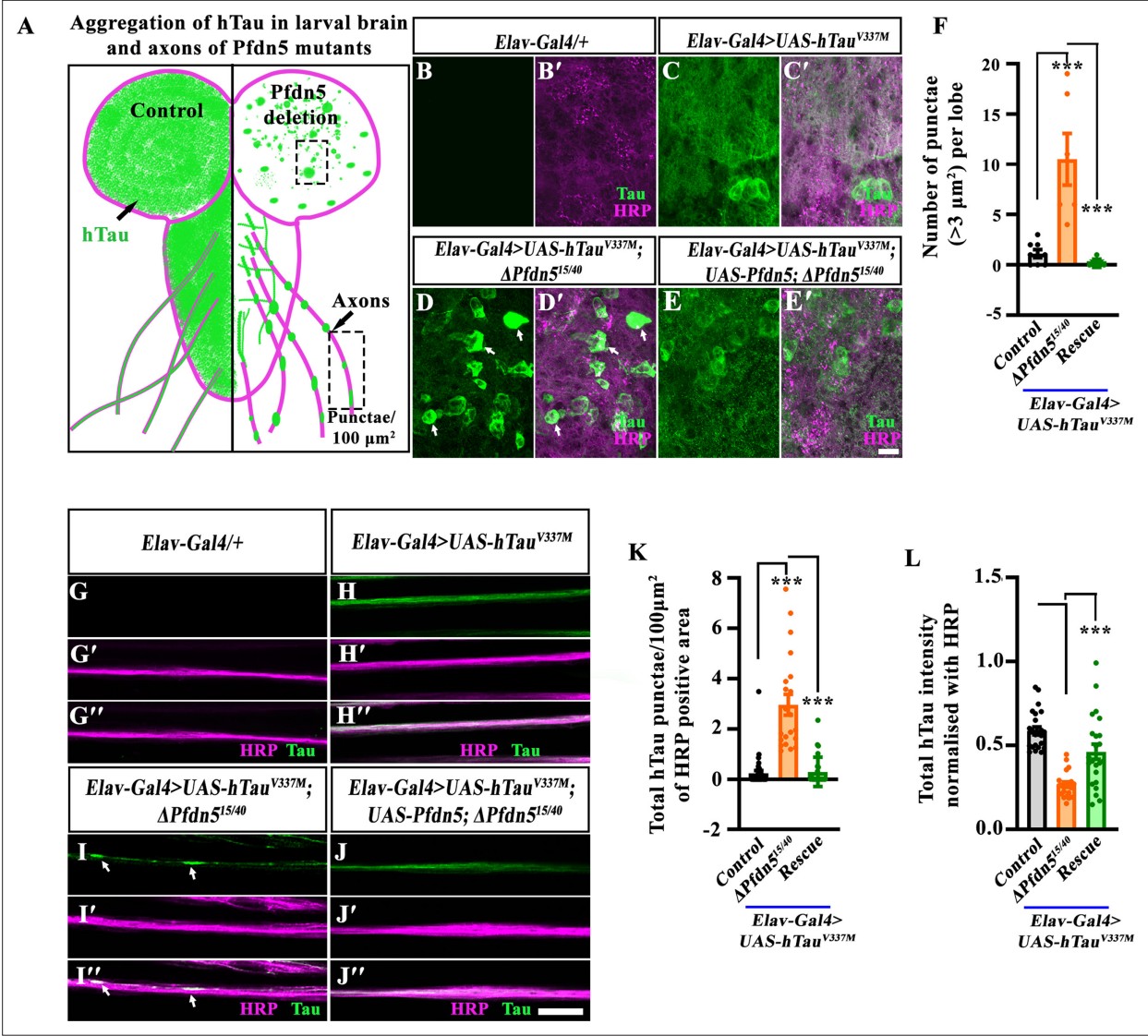

**Figure 5.** Loss of Pfdn5 induces formation of hTau[V337M] aggregates in larval neurons. (**A**) Schematic representation of pathological hTau distribution in larval brain lobes and axons of the control (left half) or *Pfdn5* mutant (right half) animals. (**B–E'**) Confocal single-section images of third instar larval brain in (**B-B'**) *Elav-Gal4/+* (control), (**C-C'**) *Elav-Gal4>UAS-hTau[V337M]*, (**D-D'**) *Elav-Gal4>UAS-hTau[V337M]; ΔPfdn5[15/40]*, (**E-E'**) *Elav-Gal4>UAS-hTau[V337M]; UAS-Pfdn5; ΔPfdn5[15/40]* double immunolabeled with neuronal membrane marker, HRP (magenta), and T46 antibody against hTau (green). The scale bar in E' for (**B-E'**) represents 10 μm. Arrows in D and D' point to the hTau punctae/flame-shaped aggregates in the brain. (**F**) Histogram showing the quantification of the number of hTau punctae (>3 μm²) in *Elav-Gal4>UAS-hTau[V337M]* (1.13±0.39), *Elav-Gal4>UAS-hTau[V337M]; ΔPfdn5[15/40]* (10.5±2.57), and *Elav-Gal4>UAS-hTau[V337M]; UAS-Pfdn5/+; ΔPfdn5[15/40]* (0.17±0.17). ***$p<0.001$. At least six optic lobes of each genotype were used for quantification. (**G-J''**) Confocal single section images of third instar larval axons in (**G-G''**) *Elav-Gal4/+* (control), (**H-H''**) *Elav-Gal4>UAS-hTau[V337M]*, (**I-I''**) *Elav-Gal4>UAS-hTau[V337M]; ΔPfdn5[15/40]* (**J-J''**) *Elav-Gal4>UAS-hTau[V337M]; UAS-Pfdn5; ΔPfdn5[15/40]* double immunolabeled for HRP (magenta), and T46 antibody against hTau (green). The scale bar in J'' for (**G-J''**) represents 10 μm. Arrows in I and I'' point to the hTau[V337M] aggregates in axons. (**K**) Histogram showing the quantification of the number of hTau punctae normalized with HRP positive area in *Elav-Gal4>UAS-hTau[V337M]* (0.25±0.1), *Elav-Gal4>UAS-hTau[V337M]; ΔPfdn5[15/40]* (2.96±0.4), and *Elav-Gal4>UAS-hTau[V337M]; UAS-Pfdn5/+; ΔPfdn5[15/40]* (0.29±0.1). ***$p<0.001$. At least 20 axons from eight animals of each genotype were used for quantification. (**L**) Histogram showing the intensity of total hTau normalized with HRP in *Elav-Gal4/UAS-hTau[V337M]* (0.59±0.02), *Elav-Gal4>UAS-hTau[V337M]; ΔPfdn5[15/40]* (0.27±0.02), and *Elav-Gal4>UAS-hTau[V337M]; UAS-Pfdn5/+; ΔPfdn5[15/40]* (0.46±0.4). ***$p<0.001$. At least 20 axons from eight animals of each genotype were used for quantification.

The online version of this article includes the following source data and figure supplement(s) for figure 5:

**Source data 1.** Source data related to *Figure 5*.

**Figure supplement 1.** hTau aggregates in the axons and larval brain of the *Pfdn5* mutants.

**Figure supplement 1—source data 1.** Source data related to *Figure 5—figure supplement 1*.

*Figure 5 continued on next page*

*Figure 5 continued*

**Figure supplement 2.** Loss of Pfdn5 results in the formation of stable hTau aggregates.

**Figure supplement 2—source data 1.** Source data related to *Figure 5—figure supplement 2*.

**Figure supplement 2—source data 2.** PDF files containing original Western blots for *Figure 5—figure supplement 2A*, indicating the relevant bands.

**Figure supplement 2—source data 3.** Original files for Western blot analysis displayed in *Figure 5—figure supplement 2A*.

**Figure supplement 3.** Loss of Pfdn5 induces Tau aggregation independent of hyperphosphorylation.

**Figure supplement 3—source data 1.** Source data related to *Figure 5—figure supplement 3*.

from soluble and/or microtubule-associated state to an aggregated, insoluble, and pathogenic state in a Tau-hyperphosphorylation-independent manner.

## Neuronal overexpression of Pfdn5 or Pfdn6 ameliorates the hTau-induced age-dependent progression of the neurodegeneration

The observations above indicate that loss of Pfdn5 enhances the neurotoxicity in the *Drosophila* Tauopathy model. We, therefore, further tested whether overexpression of Pfdn5 could alleviate Tau-induced developmental toxicity in the ommatidia. We examined the effects of Pfdn5/6 over-expression on hTau$^{V337M}$-induced ommatidial degeneration. *GMR-Gal4*-mediated overexpression of Pfdn5 or Pfdn6 in eyes significantly rescued hTau$^{V337M}$-induced ommatidial degeneration in flies (*Figure 6A-D'*, *Figure 6—figure supplement 1*). We found that *UAS-hTau$^{V337M}$/+; GMR-Gal4/++*flies showed 29.12±2.3% fused ommatidia and 70.96±3.00% degenerated eye area. This ommatidial degeneration was greatly suppressed when either Pfdn5 (% fused ommatidia, *UAS-hTau$^{V337M}$/+; GMR-Gal4/UAS-Pfdn5*: 2.98±0.31; $p<0.001$: % degenerated eye area, *UAS-hTau$^{V337M}$/+; GMR-Gal4/UAS-Pfdn5*: 5.30±0.94; $p<0.001$) or Pfdn6 (% fused ommatidia, *UAS-hTau$^{V337M}$/+; GMR-Gal4/UAS-Pfdn6*: 2.78±0.60; $p<0.001$: % degenerated eye area *UAS-hTau$^{V337M}$/+; GMR-Gal4/UAS-Pfdn6*: 3.83±1.37; $p<0.001$) was coexpressed with pathological hTau (*Figure 6A–F*). In order to ascertain that the suppression of ommatidial degeneration was not due to the Gal4 dilution, we co-expressed the neutral gene product GFP along with hTau$^{V337M}$. As expected, we found no change in the Tau-induced eye phenotype when expressed alone or with GFP (*Figure 6—figure supplement 2A–B*). These data suggest that the suppression of Tau-induced neurotoxicity was due to the expression of Pfdn5 or Pfdn6.

Overexpression of Pfdn5 suppressed not only hTau$^{V337M}$-induced neurotoxicity but also in a different Tauopathy model, hTau$^{R406W}$, which causes more severe neurotoxicity than hTau$^{V337M}$ in the fly compound eye (degenerated eye area: *GMR-Gal4 >UAS-hTau$^{R406W}$*: 82.15±3.194 vs. *GMR-Gal4 >UAS-hTau$^{R406W}$*; *UAS-Pfdn5*: 63.11±3.49) (*Figure 6—figure supplement 2C–E*), indicating that Pfdn5 can mitigate the neurodegeneration caused by at least one another variant/structural conformation of hTau. Moreover, neuronal expression of Pfdn5 and Tau$^{WT}$ or Tau$^{V337M}$ rescues the synaptic defects (*Figure 6—figure supplement 2F–L*), suggesting that Pfdn5 can alleviate multiple forms of Tauopathy.

A key pathological feature of Tau-induced neurodegeneration is age-dependent brain vacuolization in *Drosophila* and brain atrophy in humans, both of which are neuropathological hallmarks directly indicative of neuronal loss. Brain vacuolization is observed in *Drosophila* models of Tauopathy (*Byrns et al., 2021*). We, therefore, tested whether elevating the expression level of Pfdn5 or Pfdn6 could mitigate the appearance of vacuoles that can be detected in the brains of 30-day-old flies expressing hTau (*Wittmann et al., 2001*). We neuronally co-expressed hTau$^{V337M}$ alone or together with Pfdn5 or Pfdn6 using *Elav-Gal4* and examined whole-mount brain preparations stained with rhodamine-phalloidin using confocal microscopy (*Behnke et al., 2021*). Consistent with the previous reports, we found several large vacuoles in the 30 day flies expressing hTau$^{V337M}$ compared to the 2 day or 14-day-old brains (*Figure 6H-J*, *Figure 6—figure supplement 3*). The average number and size of the vacuoles in *Elav-Gal4 >UAS-hTau$^{V337M}$* (75.17±10.47 vacuoles/ brain, and 50.06±11.52 μm average vacuole size) were far higher compared to the control *Elav-Gal4/+* (7.13±0.58 vacuoles/ brain, and 5.9±0.92 μm average vacuole size) (*Figure 6M–N*). Coexpression of Pfdn5 or Pfdn6 with hTau$^{V337M}$ significantly reduced the number of vacuoles in *Elav-Gal4 >UAS-hTau$^{V337M}$* flies. Indeed, vacuole numbers and size were restored to near control levels in *Elav-Gal4 >UAS-hTau$^{V337M}$; UAS-Pfdn5/+* (7.86±0.91 and 6.76±1.57 μm) and *Elav-Gal4 >UAS-hTau$^{V337M}$; UAS-Pfdn6/++* flies (5.67±1.69 and 4.36±1.08 μm), respectively (*Figure 6K–N*). Next, we asked whether the rescue of Tau phenotypes by Pfdn5 was due to enhanced degradation of the Tau protein in the brain. Western blot analysis revealed no significant

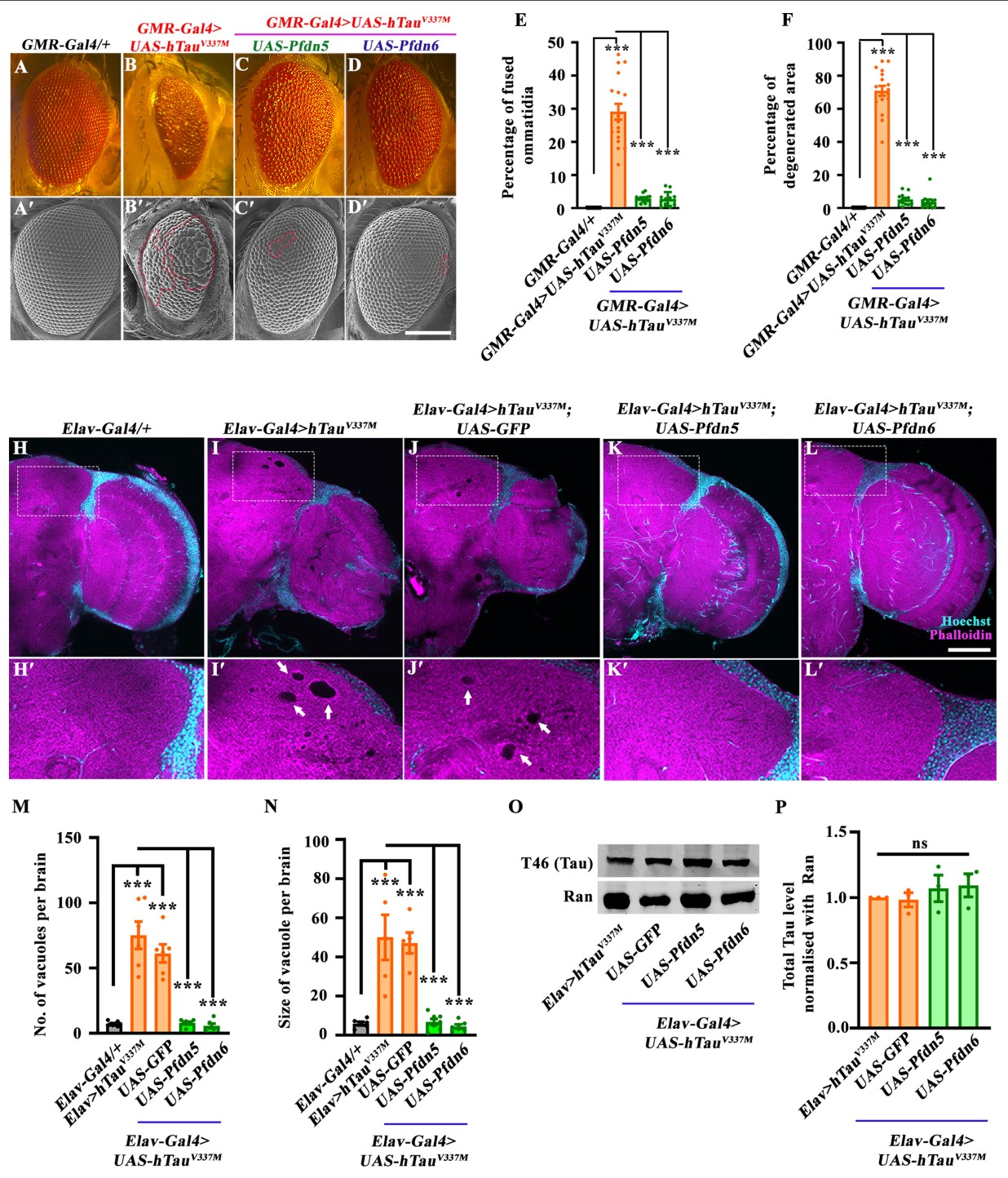

**Figure 6.** Overexpression of Pfdn5 or Pfdn6 suppresses age-dependent progression of hTau-induced neurodegeneration. (**A–D**) Bright-field images of *Drosophila* eyes expressing (**A**) *GMR-Gal4/+* (control), (**B**) *GMR-Gal4>UAS-hTau^V337M^*, (**C**) *GMR-Gal4>UAS-hTau^V337M^; UAS-Pfdn5*, (**D**) *GMR-Gal4>UAS-hTau^V337M^; UAS-Pfdn6*. (**A'–D'**) Scanning electron microscopic images of *Drosophila* eyes expressing (**A'**) *GMR-Gal4/+* (control), (**B'**) *GMR-Gal4>UAS-hTau^V337M^*, (**C'**) *GMR-Gal4>UAS-hTau^V337M^; UAS-Pfdn5*, (**D'**) *GMR-Gal4>UAS-hTau^V337M^; UAS-Pfdn6*. (**E**) Histogram showing the percentage of fused ommatidia in *GMR-Gal4/+* (0.00±0.00), *UAS-hTau^V337M^/+; GMR-Gal4/+* (29.12±2.37), *UAS-hTau^V337M^/+; GMR-Gal4/UAS-Pfdn5* (2.98±0.31), *UAS-hTau^V337M^/+; GMR-Gal4/UAS-Pfdn6* (2.78±0.60). ***p<0.001. At least 12 SEM eye images of each genotype were used for quantification. (**F**) Histogram showing percentage of degenerated area in *GMR-Gal4/+* (0.00±0.00), *UAS-hTau^V337M^/+; GMR-Gal4/+* (70.96±3.00), *UAS-hTau^V337M^/+; GMR-Gal4/ UAS-Pfdn5* (5.30±0.94), *UAS-hTau^V337M^/+; GMR-Gal4/UAS-Pfdn6* (3.83±1.37). ***p<0.001. At least 12 SEM eye images of each genotype were used for quantification. (**H–L**) Confocal images of a single section of 30-day-old adult brain in (**H**) *Elav-Gal4/+* (control), (**I**) *Elav-Gal4>UAS-hTau^V337M^*, (**J**) *Elav-*

*Figure 6 continued on next page*

*Figure 6 continued*

*Gal4>hTau^{V337M}; UAS-GFP*, (**K**) *Elav-Gal4>hTauV^{V337M}; UAS-Pfdn5*, (**L**) *Elav-Gal4>hTauV^{V337M}; UAS-Pfdn6* double immunolabeled with Hoechst (cyan), and Phalloidin (magenta). The insets represent the 3 x magnified portion of the image. Arrows point to the pathological vacuolar structures. The scale bar in L for (**H–L**) represents 20 μm. (**M**) Histogram showing the quantification of number of vacuoles in 30-day-old adult brain in *Elav-Gal4/+* (7.13±0.58), *Elav-Gal4>UAS-hTau^{V337M}* (75.17±10.47), *Elav-Gal4>hTauV^{V337M}; UAS-GFP/+* (61.17±6.91), *Elav-Gal4>hTauV^{V337M}; UAS-Pfdn5/+* (7.86±0.91), *Elav-Gal4>hTauV^{V337M}; UAS-Pfdn6/+* (5.67±1.69). ***p<0.001. At least six brains of each genotype were used for quantification. (**N**) Histogram showing the quantification of vacuole size (in μm²) in 30-day-old adult brain in *Elav-Gal4/+* (5.9±0.92), *Elav-Gal4>UAS-hTau^{V337M}* (50.06±11.52), *Elav-Gal4>hTauV^{V337M}; UAS-GFP/+* (47.17±5.39), *Elav-Gal4>hTauV^{V337M}; UAS-Pfdn5/+*(6.76±1.57), *Elav-Gal4>hTauV^{V337M}; UAS-Pfdn6/+* (4.36±1.08). ***p<0.001. At least 6 brains of each genotype were used for quantification. (**O**) Western blot showing protein levels of total Tau in *Elav-Gal4>UAS-hTau^{V337M}, Elav-Gal4>hTauV^{V337M}; UAS-GFP* (Gal4-dilution control), *Elav-Gal4>hTauV^{V337M}; UAS-Pfdn5, Elav-Gal4>hTauV^{V337M}; UAS-Pfdn6*. Ran protein levels were used as an internal loading control.

The online version of this article includes the following source data and figure supplement(s) for figure 6:

**Source data 1.** Source data related to *Figure 6*.

**Source data 2.** PDF files containing original Western blots for *Figure 6O*, indicating the relevant bands.

**Source data 3.** Original files for Western blot analysis displayed in *Figure 6O*.

**Figure supplement 1.** Pfdn5 rescues progressive eye degeneration induced by expression of Tau^{V337M}.

**Figure supplement 1—source data 1.** Source data related to *Figure 6—figure supplement 1*.

**Figure supplement 2.** Coexpression of Pfdn5 with hTau variants rescues the ommatidial degeneration and synaptic defects.

**Figure supplement 2—source data 1.** Source data related to *Figure 6—figure supplement 2*.

**Figure supplement 3.** Overexpression of Pfdn5 suppresses the age-dependent vacuolization in hTau^{V337M}-expressing flies.

**Figure supplement 3—source data 1.** Source data related to *Figure 6—figure supplement 3*.

change in the Tau levels in Pfdn5 or Pfdn6 rescue animals when compared to those expressing Tau alone (*Figure 6O*), suggesting that Pfdn5 mitigates Tau aggregates by stabilizing the microtubules rather than protein degradation. Altogether, these data indicate that increased expression of Pfdn5 or Pfdn6 can remarkably counteract neuronal loss and delay the onset and progression of the neuro-degenerative cascade induced in Tauopathy through a mechanism that involves Pfdn-mediated micro-tubule stabilization.

## Expression of Pfdn5 or Pfdn6 suppresses Tau-induced memory impairment

Cognitive decline is a common preclinical and early feature of Tauopathies (*Hanseeuw et al., 2019*). Hence, we further examined whether Pfdn5 or Pfdn6 overexpression could rescue cognitive, memory, and behavioural deficits caused by hTau^{V337M} in the *Drosophila* brain (*Orr et al., 2017*). To test the memory impairments, we used a recently developed method to assess long-term aversive olfactory conditioning memory (*Mohandasan et al., 2022*). In this method, flies learn to associate bitter food (CuSO₄) with an odor (2,3-butanedione (2,3 BD)) over 8 training cycles. Memory is assessed 24 hr later as avoidance of 2,3-BD in a Y-maze (*Figure 7A*). Untrained flies responded normally to the odor, while trained flies avoided it, indicating proper memory performance towards the conditioned odorant. Control flies (*UAS-hTau^{V337M}/+*) showed normal memory (*UAS-hTau^{V337M}/+*: naïve, 23.66±2.42 v/s trained, 10.4±2.14; *p<0.001*) (*Figure 7B*). However, animals expressing hTau^{V337M} in neurons (*Elav-Gal4 >UAS-hTau^{V337M}*: naïve, 28.39±3.47 v/s trained, 29.13±4.65; *p=0.90*) showed no difference between naïve and trained genotype (*Figure 7C*).

Next, we assessed whether the expression of Prefoldins impacted the memory deficit phenotype of flies expressing hTau^{V337M}. We first examined the effect of expression of Pfdn5 or Pfdn6 on memory performance. We found that pan-neuronal expression of either Pfdn5 (*Elav-Gal4/+; UAS-Pfdn5/+*: naïve, 18.27±2.75 v/s trained, –4.43±3.21; *p<0.001*) or Pfdn6 (*Elav-Gal4/+; UAS-Pfdn6/+*: naïve, 18.37±3.19 v/s trained, –0.88±4.73; *p<0.001*) does not cause any defect in naive odor response or memory performance after training (*Figure 7D–E*). However, coexpression of Pfdn5 significantly rescued the hTau^{V337M}-induced memory defects (*Elav-Gal4/UAS-hTau^{V337M}; UAS-Pfdn5/+*: naïve, 14.28±1.96 v/s trained, –1.5±1.7; *p<0.001*) (*Figure 7F*). Similarly, co-expression of Pfdn6 also significantly restored the hTau^{V337M}-induced memory defects (*Elav-Gal4/UAS-hTau^{V337M}; UAS-Pfdn6/+*: naïve, 20.73±4.58 v/s trained, –0.52±5.07; *p<0.001*) (*Figure 7G*). Together, these data strengthen our

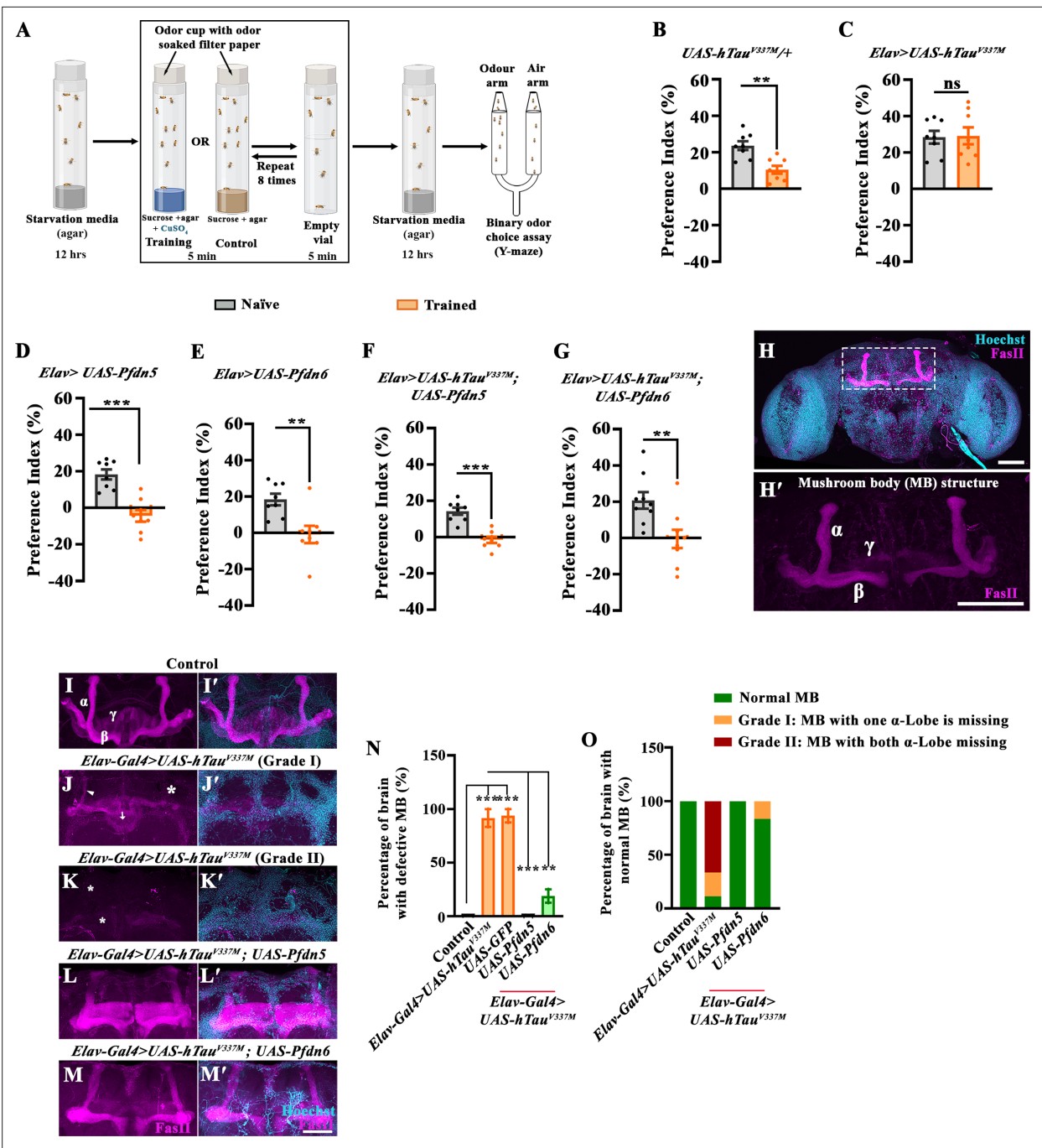

**Figure 7.** Overexpression of Pfdn5 or Pfdn6 rescues Tubulin-associated unit (Tau)-induced defects in learning and memory. (**A**) Cartoon of the Y-maze assay used for behavioural testing of conditioned odor preferences. Schematics of the protocol used to induce and measure a form of learning and memory (***Mohandasan et al., 2022***). During training, flies are exposed to a normally attractive odorant 2,3-butanedione (10⁻³-fold dilution) in a spaced training protocol: 8 X repeats of a training trial involving 5 min in the presence of unpleasant medium (80 mM CuSO₄ +85 mM sucrose and 0.75% agar) followed by 5 min in an air-filled empty vial. Trained flies were tested in a binary odor-choice assay in a Y-maze apparatus for their odor vs. air preference. Control flies were trained in 0.75% agar media with 85 mM sucrose and similarly tested in Y-maze. (**B–G**) Histogram showing the quantification of odor preference index of naïve and trained flies towards 2,3-BD in Y-maze showing normal memory in control. A reduction in the preference index after training reflects levels of learning and memory (**B**) *UAS-hTau^V337M/+*; naïve flies (23.66±2.42), trained flies (10.4±2.14), whereas pan-neuronal expression of the pathological variant hTau causes a defect in long-term memory response (**C**) *Elav-Gal4/UAS-hTau^V337M*; naïve flies (28.39±3.47), trained flies (29.13±4.65). Notably, pan-neuronal overexpression of Pfdn5 (**D**) *Elav-Gal4/+; UAS-Pfdn5/+*; naïve flies (18.27±2.75), trained flies (–4.43±3.21), pan-neuronal overexpression of Pfdn6 (**E**) *Elav-Gal4/+; UAS-Pfdn6/+*; naïve flies (18.37±3.19), trained flies (–0.88±4.73) were normal. Interestingly, pan-neuronal overexpression of Pfdn5 along with hTau^V337M expression rescues the learning and memory deficits in (**F**) *Elav-Gal4/UAS-hTau^V337M; UAS-Pfdn5/+*;

*Figure 7 continued on next page*

*Figure 7 continued*

naïve flies (14.28±1.96), trained flies (–1.5±1.7). Consistently, pan-neuronal over-expression of Pfdn6 along with hTau$^{V337M}$ expression also rescues the learning and memory deficits in (**G**) *Elav-Gal4/UAS-hTau$^{V337M}$; UAS-Pfdn6/+;* naïve flies (20.73±4.58), trained flies (–0.52±5.07). n=8 biological replicates in each case. Error bars represent the standard error of the mean (SEM). \*\*p<0.01; \*\*\*p<0.001; ns, not significant. (**H**) Confocal image of *Drosophila* brain labeled with anti-FasII antibody (magenta) and Hoechst (cyan) showing mushroom body structure in wild-type animals. Scale bar 50 μm. (**H'**) Confocal image of *Drosophila* mushroom body showing α-lobe, β-lobe, and γ-lobe. (**I-M'**) Confocal images showing mushroom body organization in (**I-I'**) control, (**J-J'**) *Elav-Gal4>UAS-hTau$^{V337M}$* Grade I (Grade I represents the defective mushroom body with one α-lobe missing), (**K-K'**) *Elav-Gal4>UAS-hTau$^{V337M}$* Grade II (Grade II represents the severe defects in mushroom body with both α-lobe missing), (**L-L'**) *Elav-Gal4>UAS-hTau$^{V337M}$; UAS-Pfdn5,* (**M-M'**) *Elav-Gal4>UAS-hTau$^{V337M}$; UAS-Pfdn6* double immunolabeled with FasII (magenta) and Hoechst (cyan). Scale bar in (**M'**) for (**I-M'**) represents 20 μm. The arrow points towards the β-lobe crossing midline, the arrowhead points to the thinner α lobe, and the asterisk represents the missing lobe. (**N**) Histogram showing the quantification for the percentage of *Drosophila* brain having defective mushroom body in control (0.00±0.00), *Elav-Gal4>UAS-hTau$^{V337M}$* (91.67±8.33), *Elav-Gal4>UAS-hTau$^{V337M}$; UAS-GFP* (93.75±6.25), *Elav-Gal4>UAS-hTau$^{V337M}$; UAS-Pfdn5* (0.00±0.00), *Elav-Gal4>UAS-hTau$^{V337M}$; UAS-Pfdn6* (18.75±6.25). \*\*\*p<0.001; ns, not significant. The error bar represents the standard error of the mean (SEM); the statistical analysis was done using one-way ANOVA. (**O**) Histogram showing the quantification of mushroom body organization as normal mushroom bodies (NBs), Grade I, or Grade II. In controls (NBs: 100, Grade I: 0, Grade II: 0), *Elav-Gal4>UAS-hTau$^{V337M}$* (NBs: 11.1, Grade I: 22.2%, Grade II: 66.7), *Elav-Gal4>UAS-hTau$^{V337M}$; UAS-Pfdn5* (NBs: 100, Grade I: 0, Grade II: 0), *Elav-Gal4>UAS-hTau$^{V337M}$; UAS-Pfdn6* (NBs: 83.3, Grade I: 16.6, Grade II: 0).

The online version of this article includes the following source data for figure 7:

**Source data 1.** Source data related to *Figure 7*.

observations that neuronal expression of Pfdn5 or Pfdn6 not only rescues Tau-induced neurodegeneration but also learning and memory deficits.

In *Drosophila,* the mushroom body is crucial for associative learning and memory. Mushroom body neuroblasts (MBNBs) produce Kenyon cells, which differentiate into three subtypes: α/β neurons, α'/β' neurons, and γ neurons (*Kunz et al., 2012*; *Figure 7H–H'*). Expression of hTau in *Drosophila* has been shown to disrupt MB architecture, predominantly affecting the α-lobe (*Mershin et al., 2004*; *Kosmidis et al., 2010*). Given that overexpression of Pfdn5 or Pfdn6 rescues the learning and memory impairments associated with hTau$^{V337M}$ expression, we examined whether Pfdn5 or Pfdn6 could prevent MB structural disruption. Immunostaining with FasII revealed severe MB abnormalities upon hTau$^{V337M}$ expression (*Figure 7I–K'*). Strikingly, coexpression of Pfdn5 or Pfdn6 completely restored MB integrity, effectively suppressing Tau-induced toxicity (*Figure 7I–O*). These findings indicate that Pfdn5 and Pfdn6 protect against Tauopathy-associated memory loss by maintaining the structural integrity of the mushroom body.

## Cotranslational functions of Pfdn5 do not completely explain its effects on neuronal microtubule stability, synapse morphology, and Tau aggregation

Consistent with cell culture and biochemical studies (*Tahmaz et al., 2021*; *Gestaut et al., 2022*), *Drosophila* Pfdn5 regulates tubulin monomers essential for microtubule assembly. Thus, one mechanism by which Pfdn5 influences Tau could be via its effect on tubulin levels. However, since Pfdn5 colocalizes and binds neuronal microtubules, it could, alternatively, directly stabilize microtubules in axons and, by allowing Tau association with microtubules, prevent aggregation of free cytoplasmic Tau protein. To examine these models, we increased tubulin monomer levels in *Pfdn5* mutants by neuronally expressing *α-Tub* transgene and asked if it restored tubulin levels in the fly, and whether such restoration would be sufficient to rescue the neuronal microtubules and synaptic defects observed in *Pfdn5* mutants. Neuronal expression of *α-Tub* in *Pfdn5* mutant background restored both α- and β-tubulin monomers as well as ace-Tubulin to near wild-type levels (ace-Tubulin level: *ΔPfdn5$^{15/40}$* (0.34±0.13) vs *Elav-Gal4 >UAS-α-Tubulin; ΔPfdn5$^{15/40}$* (1.38±0.15); p<0.01) (*Figure 8A*, *Figure 8— figure supplement 1A-D*). However, this was insufficient to rescue the axonal microtubule level and organization (Tubulin intensity at synapses: *ΔPfdn5$^{15/40}$* (0.18±0.01), *Elav-Gal4 >UAS-α-Tubulin; ΔPfdn5$^{15/40}$* (0.23±0.01); p>0.24) (*Figure 8B–E*) or synaptic phenotypes associated with *Pfdn5* mutations (satellite boutons: *ΔPfdn5$^{15/40}$* (15.5±1.48), *Elav-Gal4 >UAS-α-Tubulin; ΔPfdn5$^{15/40}$* (14.25±1.39); p>0.65) (*Figure 8—figure supplement 1E–I*). Moreover, axonal hTau aggregates seen in neuronally expressing hTau$^{V337M}$ in Pfdn5 mutants were not reduced when tubulin monomer levels were restored (hTau punctae: *Elav-Gal4/UAS-hTau$^{V337M}$; ΔPfdn5$^{15/40}$*: 3.32±0.65/100 μm$^2$; vs *Elav-Gal4/UAS-hTau$^{V337M}$; UAS-α-tubulin, ΔPfdn5$^{15/40}$*: 2.9±0.57/100 μm$^2$; p>0.80) (*Figure 8F–I*). These results prompted us to

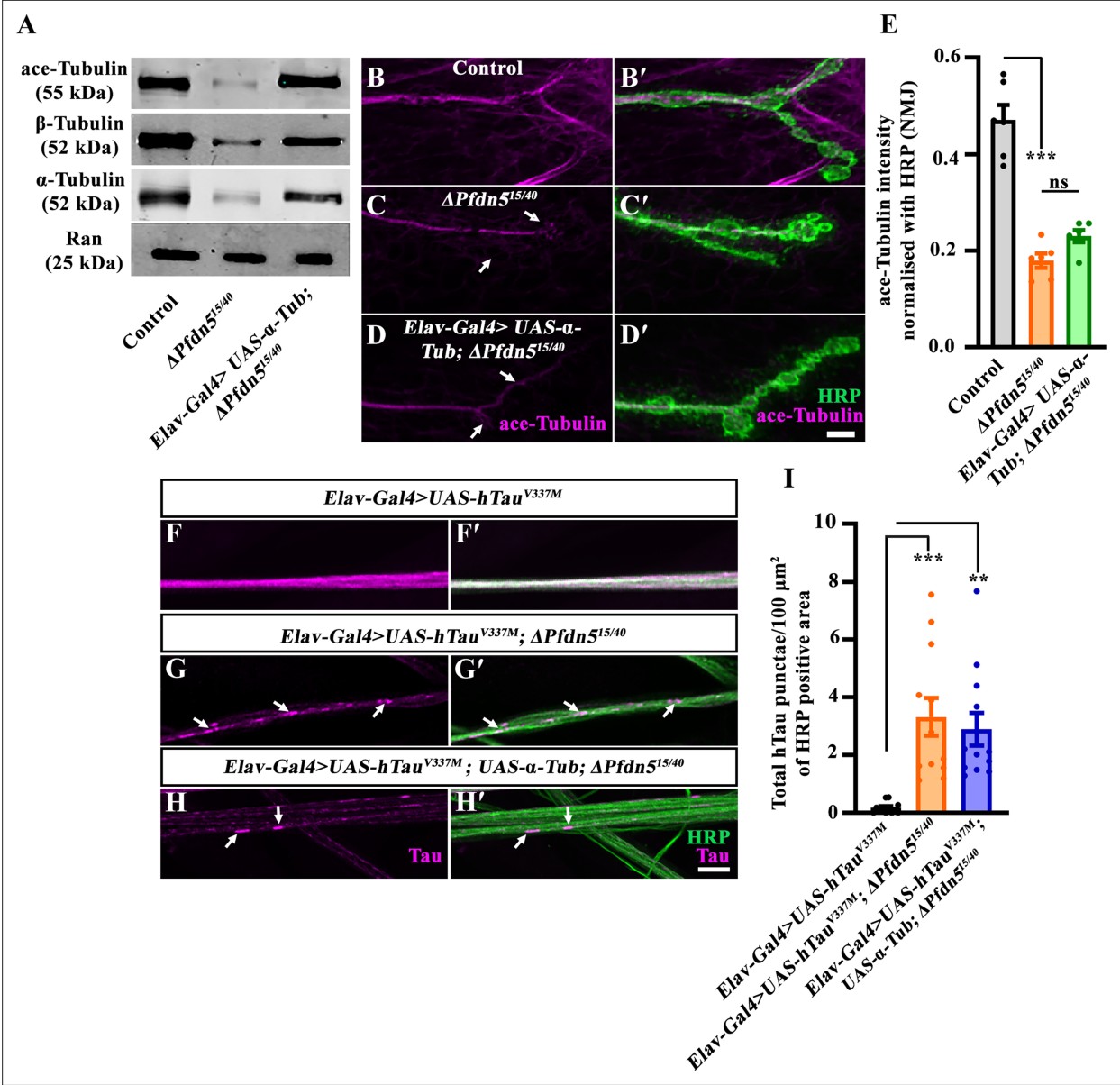

**Figure 8.** Microtubule stability and Tubulin-associated unit (Tau) association with microtubules require Pfdn5 functions downstream of tubulin monomer expression. (**A**) Western blot showing protein levels of ace-Tubulin, α-Tubulin, and β-Tubulin in control, *ΔPfdn15/40*, and *Elav-Gal4>UAS-α-Tubulin; ΔPfdn515/40*. Ran protein levels were used as an internal loading control. (**B-D'**) Confocal images of neuromuscular junction (NMJ) synapses showing synaptic microtubules in (**B-B'**) control, (**C-C'**) *ΔPfdn515/40*, (**D-D'**) *Elav-Gal4>UAS-α-Tubulin; ΔPfdn515/40* double immunolabeled for ace-Tubulin (magenta) and HRP (green). The scale bar in D' for (**A-D'**) represents 10 µm. Arrows in C and D show disrupted microtubules. (**E**) Histogram showing ace-Tubulin intensity at the NMJ in control (0.47±0.03), *ΔPfdn515/40* (0.18±0.01), *Elav-Gal4>UAS-α-Tubulin; ΔPfdn515/40* (0.23±0.01). ***$p<0.001$; ns, not significant. At least six NMJs of each genotype were used for quantification. (**F-H'**) Confocal images of third instar larval axons in (**F-F'**) *Elav-Gal4>UAS-hTauV337M*, (**G-G'**) *Elav-Gal4>UAS-hTauV337M; ΔPfdn515/40*, and (**H-H'**) *Elav-Gal4>UAS-hTauV337M; UAS-α-Tubulin, ΔPfdn515/40* double immunolabeled with neuronal membrane marker, HRP (green), and T46 antibody against total human Tau (magenta). The scale bar in H' for (**F-H'**) represents 10 µm. Arrows point to the Tau aggregates. (**I**) Histogram showing the quantification of the number of Tau punctae per 100 µm² normalized with HRP positive area in *Elav-Gal4/UAS-hTauV337M* (0.18±0.05), *Elav-Gal4/UAS-hTauV337M; ΔPfdn515/40* (3.32±0.65), and *Elav-Gal4>UAS-hTauV337M; UAS-α-Tubulin, ΔPfdn515/40* (2.9±0.57). ***$p<0.001$; **$p<0.01$. At least 12 axons from three animals of each genotype were used for quantification.

The online version of this article includes the following source data and figure supplement(s) for figure 8:

**Source data 1.** Source data related to *Figure 8*.

**Source data 2.** PDF files containing original Western blots for *Figure 8A*, indicating the relevant bands.

**Source data 3.** Original files for Western blot analysis displayed in *Figure 8A*.

*Figure 8 continued on next page*

*Figure 8 continued*

**Figure supplement 1.** Pfdn5 is required to stabilize microtubules at the synapses.

**Figure supplement 1—source data 1.** Source data related to *Figure 8—figure supplement 1*.

**Figure supplement 1—source data 2.** PDF files containing original Western blots for *Figure 8—figure supplement 1A*, indicating the relevant bands.

**Figure supplement 1—source data 3.** Original files for Western blot analysis displayed in *Figure 8—figure supplement 1A*.

examine whether Pfdn5 directly interacts with hTau. Although we observed colocalization of Pfdn5 and hTau in axons (*Figure 8—figure supplement 1J–K*), co-immunoprecipitation did not detect a physical interaction between the two proteins under the conditions we performed the pull-down experiments. Thus, these data suggest that in addition to its role as a cochaperone for tubulin monomers, Pfdn5 has an additional and potentially local role in stabilizing the neuronal microtubules as well as in preventing hTau aggregation.

## Discussion

Through varied and detailed analyses performed in established *Drosophila* Tauopathy models, we identify Prefoldin as a crucial component of chaperone systems that mitigate hTau-aggregation-induced neurodegeneration. The experiments that lead to this conclusion provide three significant insights. First, that Prefoldin acts in vivo to suppress multiple measures of Tau-mediated degeneration. Second, the mechanism of Prefoldin action in Tau-toxicity goes beyond its established role in co-translational folding of monomeric tubulin. Third, and finally, that overexpression of Prefoldin is sufficient to delay the progression of Tau-toxicity in vivo. We consider each of these issues in turn below.

### Prefoldin acts in vivo to suppress multiple measures of Tau-mediated degeneration

Seminal work by others in the field has both established the value of modeling Tauopathies in *Drosophila* and described a series of independent Tau-induced degenerative phenotypes displayed by these models (*Shulman et al., 2014*; *Zhou et al., 2017*; *Vourkou et al., 2023*; *Bukhari et al., 2024*). The initial discovery that led to the rest of our current study was the identification of subunits of the prefoldin complex in a genetic screen for modifiers of Tau-toxicity. Knockdown of Pfdn components significantly enhanced eye-ommatidial degeneration in hTau^V337M-expressing animals, suggestive of a role for this chaperone network in controlling the onset and progression of Tauopathies. Given the peripheral location of photoreceptors in the eye, it was important to more deeply assess the role of the identified chaperone components in the central nervous system. Such additional experiments confirmed that loss of Pfdn5 enhanced several additional hTau^V337M-induced phenotypes, including synaptic organization. In addition, loss of Pfdn5 resulted in a striking increase in large Tau protein aggregates in larval axons as well as in the larval brain. These data demonstrate an essential role for Pfdn in restricting hTau toxicity in vivo. However, more dramatic was the observation that neuronal overexpression of either Pfdn5 or Pfdn6 was sufficient to mitigate hTau-induced brain vacuolization and memory decline. Together, these observations demonstrate a pivotal role for Pfdn, or at least its Pfdn5 and Pfdn6 subunits, in suppressing Tau pathologies.

### Mechanism of Prefoldin action in Tau-toxicity

Molecular analysis of FTDP-17/FTLD tau mutations, as well as biochemical analysis of the pathogenic proteins, has shown that most disease-causing Tau mutations liberate Tau from microtubules and free the protein to form cytoplasmic aggregates (*Hong et al., 1998*; *Dayanandan et al., 1999*; *Guo et al., 2017*). Therefore, a simple potential mechanism by which Pfdn5 influences Tau toxicity could be through its effect on reducing levels of tubulin monomer, which would be predicted to reduce the availability of stable microtubules and thereby liberate excess Tau to form potentially pathogenic aggregates in the cytosol. Consistent with this, studies in mice and *C. elegans* have shown that neurons with reduced tubulin levels are highly susceptible to the early onset of Tau-induced neuronal dysfunction (*Tatebayashi et al., 2002*; *Yoshiyama et al., 2007*; *Miyasaka et al., 2018*). We noted that our screen for modulators of Tau-induced neurotoxicity also identified TBCE and the components of the CCT complex, which represent additional players of a chaperone network known to

participate in the cotranslational folding of nascent actin or tubulin monomers (*Hansen et al., 1999*; *Gil-Krzewska et al., 2010*; *Serna et al., 2015*). Further, our experiments showed that Pfdn5 mutations disrupt axonal microtubule organization as revealed by a reduction in the levels and organization of the microtubule-associated protein Futsch in axonal terminals of *Pfdn5* mutants (*Figure 2*; *Roos et al., 2000*; *Sherwood et al., 2004*; *Jin et al., 2009*). And finally, a prior observation that the Tau-induced eye degeneration is enhanced by the knockdown of TBCE has been proposed to be caused by perturbed microtubule dynamics in both *Drosophila* models and human patients (*Dou et al., 2003*; *Fujiwara et al., 2020*; *Battini et al., 2021*).

Despite the above observations, the effect on tubulin monomer levels does not completely explain how Pfdn influences microtubule organization or Tau toxicity. First, we report the unexpected but robust observation that Pfdn5 is a microtubule-associated protein, physically associated with stable axonal microtubules and, therefore, well positioned to directly influence microtubule stability (*Figure 3*). Second and more directly, we find that genetic restoration of α-tubulin and β-tubulin monomers, as well as acetylated tubulin levels, was not sufficient to rescue the synaptic defects observed in *Pfdn5* mutants (*Figure 8*).

Pfdn5 appears to influence Tau toxicity through a mechanism downstream of its known roles in cotranslational folding of tubulin. While this could be via its function as a novel microtubule-associated protein, an additional possibility we consider is that Pfdn, and by extension other known cotranslational chaperones, could act additionally and directly as holdases or disaggregases for Tau and/or other aggregation-prone proteins. There is considerable circumstantial evidence to indicate post-translational and direct roles for Prefoldin, as well as CCT, in preventing the aggregation of misfolded proteins. For instance, Prefoldin not only inhibits the formation of larger Htt aggregates (*Tashiro et al., 2013*) or amyloid β-aggregates (*Sörgjerd et al., 2013*) but also solubilizes the amyloid oligomers and inhibits their fibril formation under in vitro conditions (*Sakono et al., 2008*; *Sörgjerd et al., 2013*). Similarly, the CCT/TRiC complex physically associates with the polyQ repeats of Htt protein and remodels pathogenic aggregates in vitro (*Tam et al., 2006*; *Darrow et al., 2015*). This evidence supports a direct role of Prefoldin and CCT as 'disaggregase' or 'aggregate remodellar' for aggregate-prone proteins and might regulate assembly/disassembly of Tau protein.

The identification of this cytoskeleton-regulatory chaperone network as a major modulator of Tauopathy supports the hypothesis that an age-dependent compromise in the chaperone activity could vitiate the onset and progression of multiple forms of Tauopathies and potentially other neurodegenerative diseases (*Tittelmeier et al., 2020*; *Cyske et al., 2023*). Pfdn5 levels have been reported to decrease with age in a mouse model of Tauopathy *Kadoyama et al., 2019*; our findings provide direct evidence that even a minimal amount of pathological hTau in the absence of Pfdn5 could induce the early onset of hTau-induced neurodegeneration. Moreover, our finding that neuronal expression of α-tubulin rescues the hTau-induced synaptic defects in a manner that critically requires Pfdn5 activity extends the functional requirement of Prefoldins in the suppression of Tauopathies beyond their reported activity in cell culture or in vitro models (*Millán-Zambrano et al., 2013*; *Sörgjerd et al., 2013*; *Tashiro et al., 2013*; *Takano et al., 2014*).

## Prefoldin overexpression as a strategy to mitigate Tauopathies

Our work suggests that stabilizing the components of this chaperone system, particularly Pfdn5 or Pfdn6, could be a promising therapeutic approach for delaying Tau-induced neuropathology. Neuronal overexpression of Pfdn5 or Pfdn6 did not result in any detectable changes in synaptic morphogenesis or age-dependent neuronal degeneration. However, coexpression of Pfdn5 or Pfdn6 with the pathological variant of hTau remarkably suppressed Tau-induced synaptic defects, prevented brain vacuolization, and rescued memory defects in multiple forms of tauopathies (*Figures 6 and 7*). This provides clear evidence that Pfdn5/Pfdn6-dependent microtubule regulation could potentially suppress Tau-induced neurodegeneration. These conclusions are supported by prior observations that expressing an acetylation mimic form of tubulin (*Mao et al., 2017*) or stabilizing microtubules (*Xiong et al., 2013*) rescues the synaptic defects induced in the *Drosophila* Tauopathy model.

Do Prefoldins have a general neuroprotective role? In neuronal cell lines, human Prefoldins colocalize with PolyQ-expanded protein Huntingtin and prevent the formation of toxic aggregates, supporting its role in the suppression of aggregation-induced neurotoxicity (*Tashiro et al., 2013*). A recent study has shown that age-dependent microtubule defects in *Drosophila* lead to dTau aggregation similar

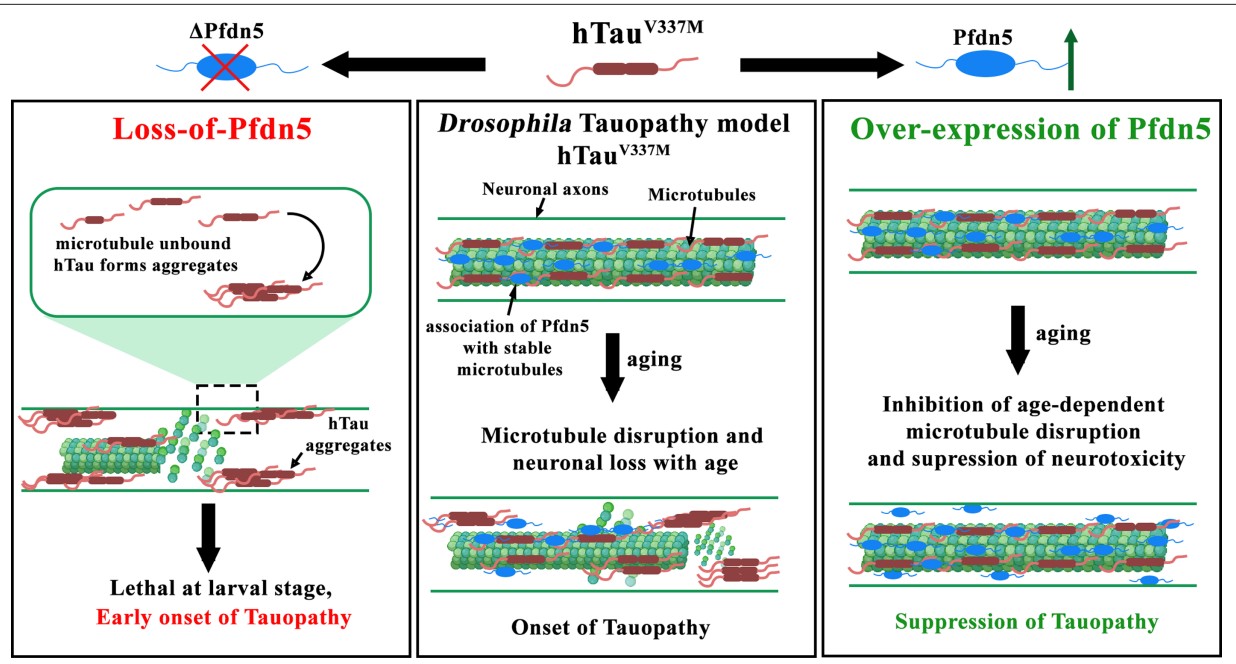

**Figure 9.** A model depicting novel functional requirements of Pfdn5 in microtubule stabilization and its role in suppressing age-dependent neuropathy. In axons, Pfdn5 physically associates with microtubules and stabilizes them, thereby suppressing the turnover of microtubules. The pathological Tau dislodges from microtubules in an age-dependent manner and forms pathological aggregates that induce neuronal death (Middle panel). Loss of Pfdn5 disrupts neuronal microtubules, resulting in abnormal synaptic morphogenesis and facilitating the dislodging of microtubule-associated Tau, resulting in the formation of Tubulin-associated unit (Tau) aggregates and stepping up the early onset of Tauopathies. An age-dependent reduction in the Pfdn5 levels or mutations in Pfdn5 could result in microtubule fragmentation and may facilitate Tau-induced neurotoxicity (Left panel). Pfdn5 suppresses Tau aggregation in a manner that involves microtubule stability and does not appear to regulate Tau solubility directly. Notably, neuronal overexpression of Pfdn5 suppresses the microtubule disruption even in aged flies, thereby inhibiting the progression of Tauopathy (Right panel).

to aged human individuals, suggesting a crucial role of microtubule stability in tauopathies (*Okenve-Ramos et al., 2024*). Our findings that loss of Pfdn5 disrupts microtubules at an early stage and leads to Tau aggregation further support the crucial requirement of Pfdn5-dependent microtubule stability in suppressing various forms of Tauopathy. In addition, several lines of supportive evidence from human Alzheimer's disease datasets implicate PFDN5 in disease pathology. For example, recent compilations and analyses of proteomic data identified CCT components, TBCE, as well as Prefoldin subunits, including PFDN5, in AD tissue (*Hsieh et al., 2019*; *Tao et al., 2020*; *Ji et al., 2022*; *Askenazi et al., 2023*; *Leitner et al., 2024*; *Sun et al., 2024*). Furthermore, whole blood mRNA expression data from Alzheimer's patients revealed downregulation of PFDN5 transcript (*Ji et al., 2022*). Together, these findings from human data support an essential role of PFDN5 in suppressing diverse neurodegenerative processes. Our data mechanistically extends these studies by revealing that Pfdn5 directly stabilizes neuronal microtubules, assists in proper partitioning of Tau onto the stable microtubules, and suppresses the formation of pathological Tau aggregates. Importantly, our data reveal that expression of Pfdn5, whether ubiquitous or neuron-specific, does not induce any observable synaptic or microtubule-associated defects in neurons. This finding holds significant therapeutic promise since modulating Pfdn5 or Pfdn6 expression or stability could safely and effectively mitigate neurodegenerative diseases associated with microtubule instability, such as Tauopathies and possibly FUS-induced neurodegeneration (*Kandhavivorn et al., 2023*).

## Concluding remarks

Based on our findings, we propose a model in which Pfdn5 regulates microtubule formation and stability by two non-exclusive mechanisms: (a) by regulating the folding of nascent tubulin monomers and (b) by directly associating and stabilizing microtubules in neurons (*Figure 9*). Both the functions of Pfdn5 are essential for regulating synaptic morphogenesis and Tau partitioning onto the microtubules. Normalizing the tubulin monomers to near wild-type levels was insufficient to rescue the

axonal microtubule organization or suppress the Tau aggregation in the absence of Pfdn5. This further supports the model that Pfdn5-dependent tubulin stabilization is essential for Tau partitioning and that Tau aggregation is microtubule-dependent. Thus, while the conventional chaperone function of Prefoldins is essential for tubulin folding, direct association of Pfdn5 with stable microtubules to limit its turnover in neurons is crucial for suppressing Tau aggregation. Further elucidation of the underlying mechanisms of Pfdn5-mediated neuroprotection and chemical screens to identify novel small molecules may pave the way for novel strategies to preserve neuronal function and combat neurodegeneration.

## Materials and methods

### Stocks and *Drosophila* husbandry

The *Drosophila* stocks were maintained at 25 °C in standard cornmeal medium containing sucrose, agar, and yeast granules. The larvae for experiments were grown at 25 °C in protein-rich media (80 g/L cornflour, 40 g/L dextrose, 20 g/L sucrose, 18 g/L agar, 15 g/L yeast extract, 4% (v/v) propionic acid, 0.06% (v/v) ortho-phosphoric acid, and 0.07% methyl-4-hydroxy benzoate/Tego) under non-crowded conditions. The $w^{1118}$ was used as a control unless otherwise stated. All the genetic combinations and recombination were made using standard *Drosophila* genetics. The crosses for RNAi-mediated knockdown and the rescue experiments were grown at 25 °C. The following *Drosophila* lines were used in this study: *UAS-hTau^{V337M}* (*Wittmann et al., 2001*), *UAS-hTau^{R406W}* (*Wittmann et al., 2001*), *UAS-α-Tub84B* (BL-7373), *actin5C-Gal4* (BL-25374); *Elav^{C155}-Gal4* (BL-458), *mef2-Gal4* (BL-50742), and *GMR-Gal4* (BL-9146). Details of RNAi lines used in this study are mentioned in *Supplementary file 1* and *Supplementary file 3*.

### Scanning electron microscopy

The flies were immersed in fixative (1% glutaraldehyde, 1% formaldehyde, and 1 M sodium cacodylate, pH 7.2) for 2 hr, followed by subsequent washes and dehydration via an ethanol series. The samples were then dried and sputter-coated as previously described (*Choudhury et al., 2016*). The flies were mounted on carbon conductive tabs stuck on aluminium stubs and imaged using a Zeiss scanning electron microscope (Carl Zeiss, Germany).

### Generation of *Pfdn5* loss-of-function mutants and Pfdn transgenes

To generate the loss-of-function mutants of Pfdn5, two sets of gRNAs were designed for the Pfdn5 genomic region using the CRISPR Optimal Target Finder online tool. The two gRNA pairs (gRNA1FP, gRNA1RP, and gRNA2FP, gRNA2RP) were cloned into a dual gRNA pCFD4 vector having a BbsI restriction site using Gibson Assembly Kit (New England Biolabs Ltd, UK) following the manufacturer's guidelines. The pCFD4 vector containing Pfdn5 gRNAs was injected into *Drosophila* embryos to generate the transgene. Next, the transgenic flies containing the Pfdn5 gRNAs were crossed with *nanos-Cas9* (BL-54591) to create the deletion of the *Pfdn5* gene in the germline cells. Following standard genetic crosses, lines were established in the F2 generation, and *Pfdn5* deletion was screened by PCR using primers Pfdn5_FP1 and Pfdn5_RP. Two null mutants of Pfdn5, *ΔPfdn5^{15}* (606 bp deletion), and *ΔPfdn5^{40}* (577 bp deletion) were obtained and verified by sequencing using primer: Pfdn5_Seq FP.

To generate Pfdn5 or Pfdn6 transgenes, a full-length Pfdn5 or Pfdn6 ORF was amplified from cDNA and cloned into the Gal4-based expression vector pUASt at EcoRI and NotI restriction sites. The pUASt vector containing the Pfdn5 or Pfdn6 ORF was injected into *Drosophila* embryos to generate the transgene. Semiquantitative RT-PCR was used to assess the expression of the *Pfdn5* transcript in *Pfdn5* mutants. In brief, total RNA was isolated from larval fillets using TRIzol reagent (Invitrogen, Waltham, MA, USA). Reverse transcription was performed on 1 µg total RNA using Superscript II Reverse Transcriptase (Invitrogen, Waltham, MA, USA) using an oligo-dT primer to make cDNA. The resulting cDNA was used for PCR to analyze the level of *Pfdn5* transcript using primers Pfdn5_RTFP and Pfdn5_RTRP. The list of primers used in this study is reported in *Supplementary file 2*.

### Generation of Pfdn5 antibody

To generate antibodies against Pfdn5, the full-length Pfdn5 was amplified from cDNA using primers Pfdn5_pET28 FP and Pfdn5_pET RP and cloned into the pET-28a (+) bacterial expression vector at

NotI and EcoRI restriction sites. The His-tagged fusion protein was expressed in BL21 codon + cells, purified from inclusion bodies using the standard protein purification method from the pellet fraction, and injected into mice (animal facility, IISER Bhopal). The antibody was used at a 1:200 dilution on fillets and a 1:5000 dilution for western blotting.

## Quantitative RT-PCR

Total RNA was isolated from larval fillets (actin-Gal4>UAS RNAi) using Qiagen RNA extraction kit, following the manufacturer's instructions (Invitrogen, Waltham, MA, USA). First-strand cDNA was synthesized using PrimeScript 1st strand cDNA Synthesis Kit (Takara, 6110 A). Quantitative RT-PCR (qRT-PCR) reactions were set up using iTaq Universal SYBR Green Supermix (#1725124, Bio-Rad) in the qTOWER³ (Analytik Jena, Jena, Germany) qPCR machine according to the manufacturer's protocol. The primers were designed using the IDT Primer Quest tool (https://www.idtdna.com) and are listed in **Supplementary file 2**. *rp49* was used as an internal control. Three independent RT-qPCR runs were performed. The fold change was calculated using $2^{-\Delta(\Delta Ct)}$ (**Livak and Schmittgen, 2001**).

## Immunocytochemistry

Wandering third instar larvae were dissected on a Sylgard plate in cold calcium-free HL3 and fixed in 4% paraformaldehyde in PBS for 30 min or in methanol for 5 min. The larval fillets were washed three times in PBS containing 0.2% Triton X-100, followed by blocking for 1 hr in 0.2% PBST containing 5% BSA. Fillets were fixed in methanol for 5 min to stain the acetubulin in the muscles. The fillets were incubated overnight at 4°C with a primary antibody, followed by fluorophore-conjugated secondary antibodies at room temperature for 90 min. Finally, larval fillets were mounted on a glass slide with Fluoromount-G aqueous mounting medium (Thermofisher, Waltham, MA, USA). Primary antibodies used in the study, mouse anti-CSP (ab49, 1:50), mouse anti-Futsch (22C10, 1:50), and mouse anti-β-tubulin (E7, 1:50) were obtained from the Developmental Studies Hybridoma Bank (University of Iowa, USA). Other primary antibodies used in this study are mouse anti-dPfdn5 (this study, 1:200), mouse anti-ace-tubulin (1:500, Sigma-Aldrich, Missouri, USA), anti-Tau (T46, 1:100, Invitrogen, Waltham, MA, USA), anti-FasII (1:50, DSHB), anti-D5D8N (1:500, CST, Boston, MA), and anti-phospho-Tau (AT8, 1:100, Invitrogen, Waltham, MA, USA). The fluorophore-conjugated secondary antibody Alexa Fluor 488 or Alexa Fluor 568 (Thermo Fisher Scientific, Waltham, MA, USA) was used at 1:800 dilution. Alexa Fluor 488 or Rhodamine-conjugated anti-HRP (Jackson ImmunoResearch, Baltimore, PA, USA) were used at 1:800 dilution. Hoechst (Thermo Fisher Scientific, Waltham, MA, USA) was used at a 1:5000 dilution for 5 min.

The brain staining for assessing vacuolization was done as previously described (**Behnke et al., 2021**). Briefly, the adult flies of appropriate genotypes were anesthetized and beheaded. The head was fixed in 4% PFA in 1 X PBS containing 0.5% Triton X-100 for 20 min. The brain was dissected and fixed for another 2 hr, washed with PBST, and incubated with Hoechst (1:5000) and Alexa Fluor 568 conjugated Phalloidin (1:100, Thermo Fisher Scientific, Waltham, MA, USA) cocktail in PBST for 24 hr. The brains were washed five times with PBST, followed by a final wash in 1 X PBS for 30 min to remove the residual detergents or air sac and mounted with Fluoromount-G aqueous mounting medium (Thermo Fisher Scientific, Waltham, MA, USA) on a glass slide for visualization.

## Western blot analysis

Third instar larval body wall muscle or adult *Drosophila* heads were homogenized in 1 X SDS lysis buffer (50 mM Tris-Cl, pH 6.8; 25 mM KCl; 2 mM EDTA; 0.3 M sucrose; 2% SDS), boiled, and centrifuged at 3000 g. The protein concentration was quantified using bicinchoninic acid (BCA) Protein assay (**Simpson, 2008**). The homogenized sample was then combined with an equal volume of 2x Laemmli buffer (50 mM Tris-HCl, pH 6.8; 2% SDS; 2% β-Mercaptoethanol; 0.1% Bromophenol blue and 10% glycerol). Subsequently, 25 µg of protein was separated on a 12% SDS-PAGE gel and transferred to a Hybond-LFP PVDF membrane (GE Healthcare, Illinois, USA). The membrane was blocked in 5% skimmed milk in 1 X Tris-buffered saline (TBS) with 0.2% Tween-20 (0.2% TBST) for 1 hr at room temperature and then incubated overnight with primary antibody. After washing with 0.2% TBST, the membrane was incubated with HRP-conjugated secondary antibody for 1 hr at room temperature. The primary antibodies used were: mouse anti-Pfdn5 (this study, 1:5000), rabbit anti-α-tubulin (1:3000, CST, Mumbai, India), mouse anti-β-tubulin (E7, 1:300, DSHB, University of Iowa, USA), mouse

anti-ace-tubulin (1:5000, Sigma-Aldrich, St. Louis, Missouri, USA), anti-Tau (T46, 1:1000, Invitrogen, Waltham, MA, USA), anti-phospho-Tau (AT8, 1:1000, Invitrogen, Waltham, MA, USA), anti-GAPDH (1:5000), and mouse anti-Ran (1:2000, BD Biosciences, New Jersey, USA). Signals were detected using the LI-COR Odyssey imaging system (LI-COR Biosciences, Lincoln, USA).

## In vivo microtubule-binding assay

The microtubule binding assay was performed as described previously (*Feuillette et al., 2010*; *Ando et al., 2016*). Fifty heads from wild-type adult flies were collected and homogenized in 100 μl of Buffer-C+ 50 mM (HEPES); pH 7.1, 1.0 mM MgCl2, 1.0 mM EGTA, protease inhibitor cocktail (Roche, Basel, Switzerland), and phosphatase inhibitor cocktail in the presence of 20 μM Taxol or 40 μM Nocodazole diluted in dimethylsulfoxide (DMSO). Homogenized heads were centrifuged at 1000×g for 10 min, and an aliquot of the supernatant was subjected to western blotting as the 'input fraction.' The remaining supernatant was layered onto a two-volume cushion of Buffer-C+ with 50% sucrose. After centrifugation at 100,000×g for 30 min, one-third of the supernatant containing soluble tubulin was collected from the top of the tube as the cytosol fraction, and the pellet containing microtubule polymers and proteins bound to microtubules was resuspended in 100 μl of SDS-Tris-Glycine sample buffer. Protein concentration in each fraction was measured using the BCA Protein Assay Kit. Equal amounts of protein were loaded onto each lane of Tris-Glycine gels and analyzed by western blotting using anti-Pfdn5 or anti-ace-tubulin antibodies.

## Tau solubility assay

Tau solubility assay was performed as described in *Vourkou et al., 2023*. Briefly, adult fly heads were homogenized in TBS/sucrose buffer (50 mM Tris HCl, pH 7.4, 175 mM NaCl, 1 M sucrose, 5 mM EDTA) supplemented with protease and phosphatase inhibitors. The homogenate was initially centrifuged at 1000×g for 2 min to remove debris. The resulting supernatant was subjected to ultracentrifugation at 200,000×g for 2 hr at 4 °C to separate soluble proteins. This supernatant comprised the soluble Tau fraction. The pellet, containing insoluble material, was resuspended in 5% SDS/TBS buffer and centrifuged again at 200,000×g for 2 hr at 25 °C. The resulting supernatant was collected as the SDS-soluble, aqueous-insoluble fraction, which was enriched for aggregated Tau. All fractions were diluted in Laemmli buffer, boiled, and resolved using SDS-PAGE. Immunoblotting with Tau-specific antibodies was used to analyze Tau species in the different fractions (*Vourkou et al., 2023*).

## Drug treatments

Third instar *Drosophila* larvae were dissected in Schneider's *Drosophila* Medium (Gibco, CA, USA). Following dissection, preparations were gently washed with fresh Schneider's medium to remove residual debris. The preparations were incubated in Schneider's medium containing 1,6-Hexanediol (1,6-HD; H6703, Sigma-Aldrich) at final concentrations of 0%, 1%, or 5% for 2 min at 25 °C. The 0% condition consisted of Schneider's medium alone (*Liu et al., 2021*). Following treatment, samples were immediately fixed in 4% paraformaldehyde in PBS for 30 min at room temperature and processed for immunostaining. For the LiCl treatments, all the genotypes were raised on the standard fly media containing zero mM or 20 mM LiCl (*Cowan et al., 2010*). The zero mM solution contained only DMSO.

## Memory paradigm for aversive associative olfactory conditioning

To induce long-term aversive conditioning memory (LTM), flies were trained to associate an attractive odorant with bitter food, CuSO₄, as described previously (*Mohandasan et al., 2022*). 4–5 day-old adult flies were trained on 0.75% agar media containing 85 mM sucrose and 80 mM CuSO₄ (punishment media); the same media without CuSO₄ was used as control media. Flies of specific genotypes were first starved in glass vials overnight containing 0.75% agar (starvation media) and then transferred to punishment media vials. For delivering the odor, a filter paper (1.5 cm × 2 cm), soaked in 100 μl of 5% 2,3 BD (2,3 butanedione, attractive odorant), was placed in a porous odor cup fitted at the top of the punishment or control vials. Starved flies were transferred into the punishment or control vials for 5 min, followed by 5 min of incubation in an empty test tube. This training cycle was repeated eight times for both the punishment and control groups of flies. For checking 1 day memory retention, flies are starved for 6 hr after the 8-cycle conditioning step, followed by a 5 min food pulse and again starvation for 18 hr. The flies were then tested (24 hr after training) for their preference

towards 2,3 BD in a binary odor choice assay paradigm using a Y-maze. The Preference Index (PI) of the control and trained flies was calculated below.

$$\text{Preference Index} = \frac{\textit{Odor Arm Flies} - \textit{Air Arm Flies}}{\textit{Odor Arm Flies} + \textit{Air Arm Flies}} X100$$

## Quantifications and statistical analysis

For bouton quantification, images were captured with a laser scanning confocal microscope (FV3000; Olympus) using 40x 1.3 NA or 60x 1.42 NA objectives and processed using ImageJ (National Institutes of Health, USA) or Adobe Photoshop software (Adobe Inc, USA). NMJs from muscle 4 at A2 hemisegment were captured using a 60x 1.42 NA objective to calculate the bouton number. CSP-positive boutons were counted manually. For bouton area quantification, NMJs from muscle 4 at A2 hemisegment were captured, and the area of five terminal boutons was calculated by drawing a free-hand sketch around CSP-positive boutons. The control and experimental fillets were processed similarly for fluorescence quantification, and the fluorescence images were captured under the same settings for every experimental set. For quantification of AT8 and T46 levels in the larval axons, HRP-marked boundaries were defined for each axon. The fluorescence intensity of AT8 or T46 was calculated and normalized with HRP fluorescence. To quantify the Tau punctae in the axons, z-projections of confocal images of third-instar larval axons were captured. T46 and AT8 positive punctae were manually counted and normalized with the area of respective axons. To quantify the Tau punctae in the larval brain, fluorescence threshold was set and analyzed using ImageJ. Tau punctae greater than >3 μm$^2$ were quantified.

For bright field imaging of eyes, flies were anesthetized using diethyl ether (Sigma-Aldrich, Missouri, USA), and images were captured using Leica M205FA (Leica, Germany) Stereo Zoom Microscope. The percentage of the degenerated area was quantified as the area of the eyes showing roughness (for bright-field images) and the area containing fused ommatidia (for SEM images), normalized with the total area of the eye multiplied by 100. The percentage of fused ommatidia was quantified from the SEM images as the number of fused ommatidia normalized with the total ommatidia multiplied by 100. The maximum area of individual vacuoles was defined using the Wand tool in ImageJ software to quantify the vacuole size. Subsequently, the traced vacuoles were assigned and saved as regions of interest (ROIs). The selected ROIs were stacked and measured to quantify the size of the vacuoles (*Behnke et al., 2021*). The total number of boutons with Futsch-positive loops was quantified manually using ImageJ (*Coyne et al., 2014*).

Colocalization analysis was performed using the JACoP ImageJ plugin (*Bolte and Cordelières, 2006*). A line was drawn across the axons, and plot profiles were drawn using the ImageJ function Plot Profile. The density of Western blot bands was quantified using ImageJ software. For multiple comparisons, one-way ANOVA followed by post hoc Tukey's test or Student's t-test was used. GraphPad Prism 8 (GraphPad Software Inc, California, USA) was used to plot all the graphs. Error bars in all the histograms represent + SEM. $*p<0.05$, $**p<0.01$, $***p<0.001$.

## Acknowledgements

We thank Drs. Mel Feany and Surajit Sarkar, the Bloomington Drosophila Stock Center (BDSC) and Vienna Drosophila Resource Centre (VDRC) for the fly stocks, the Developmental Studies Hybridoma Bank (DSHB), the University of Iowa for monoclonal antibodies, and Varun Chaudhary, Baskar Bakthavachalu, Sunando Datta, and Sankar Jha for their inputs on this manuscript. We acknowledge the DST-FIST (Government of India) supported confocal facility at IISER Bhopal. We acknowledge help from Debasis Nayak in generating the Pfdn5 antibody at IISER Bhopal animal facility. This work was supported by a research grant from the Science and Engineering Research Board (SERB Project No- EMR/2016/004718), the Government of India and intramural funds from IISER Bhopal to VK Anjali, who acknowledges fellowship support from the University Grants Commission, Government of India. MR acknowledges support from a Wellcome Trust- HRB-SFI Investigator grant, a Science Foundation Ireland Future Frontiers Programme grant, and an ANRF VAJRA grant from the Government of India.

# Additional information

## Competing interests

Mani Ramaswami: Reviewing editor, *eLife*. The other authors declare that no competing interests exist.

## Funding

| Funder | Grant reference number | Author |
|---|---|---|
| Science and Engineering Research Board | EMR/2016/004718 | Vimlesh Kumar |
| Indian Institute of Science Education and Research, Bhopal | | Anjali Bisht Vimlesh Kumar |
| Wellcome Trust | | Mani Ramaswami |
| Science Foundation Ireland Future Frontiers Programme | | Mani Ramaswami |
| ANRF VAJRA | | Mani Ramaswami |

The funders had no role in study design, data collection and interpretation, or the decision to submit the work for publication. For the purpose of Open Access, the authors have applied a CC BY public copyright license to any Author Accepted Manuscript version arising from this submission.

## Author contributions

Anjali Bisht, Conceptualization, Data curation, Software, Formal analysis, Validation, Investigation, Visualization, Methodology, Writing – original draft, Writing – review and editing; Srikanth Pippadpally, Conceptualization, Software, Investigation, Visualization, Methodology, Writing – review and editing; Snehasis Majumder, Athulya T Gopi, Investigation, Methodology; Abhijit Das, Methodology, Writing – original draft; Chandan Sahi, Conceptualization, Resources, Methodology, Writing – original draft; Mani Ramaswami, Conceptualization, Supervision, Methodology, Writing – original draft, Writing – review and editing; Vimlesh Kumar, Conceptualization, Supervision, Funding acquisition, Validation, Writing – original draft, Project administration, Writing – review and editing

## Author ORCIDs

Anjali Bisht ⓘ https://orcid.org/0000-0001-5519-4664
Srikanth Pippadpally ⓘ https://orcid.org/0000-0002-0636-9891
Mani Ramaswami ⓘ https://orcid.org/0000-0001-7631-0468
Vimlesh Kumar ⓘ https://orcid.org/0000-0003-2206-4905

Reviewer #1 (Public review): https://doi.org/10.7554/eLife.104691.3.sa1
Reviewer #2 (Public review): https://doi.org/10.7554/eLife.104691.3.sa2
Author response https://doi.org/10.7554/eLife.104691.3.sa3

# Additional files

## Supplementary files

Supplementary file 1. The table shows the list of HSPs used to screen as genetic modifiers of Tauopathies.

Supplementary file 2. Table shows the list of primers used in this study.

Supplementary file 3. Table shows details of RNAi lines used against cytoskeletal chaperones.

MDAR checklist

## Data availability

All data associated with this study is included within the manuscript.

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
