## [Editor Report · eLife Assessment]

This work details the finding that in at least one of the subunits of the heterohexameric chaperone complex Pfdn5 has additional functions beyond its contribution to cytoskeletal protein folding in Drosophila. The authors provide **convincing** evidence that it is a hitherto unknown microtubule associated protein in addition to regulating microtubule organization and levels of tubulin monomers. The **important** findings show that Pfdn5 loss exaggerates pathological manifestations of mutant human Tau bearing FTDP-17 linked mutations in Drosophila, while its overexpression suppresses them, suggesting that the latter may constitute a future therapeutic approach.

---

## [Referee Report · Reviewer #1 (Public review)]

Summary:

In this manuscript, Bisht et al address the hypothesis that protein folding chaperones may be implicated in aggregopathies and in particular Tau aggregation, as a means to identify novel therapeutic routes for these largely neurodegenerative conditions.

The authors conducted a genetic screen in the Drosophila eye, which facilitates identification of mutations that either enhance or suppress a visible disturbance in the nearly crystalline organization of the compound eye. They screened by RNA-interference all 64 known Drosophila chaperones and revealed that mutations in 20 of them exaggerate the Tau-dependent phenotype, while 15 ameliorated it. The enhancer of degeneration group included 2 subunits of the typically heterohexameric prefoldin complex and other co-translational chaperones.

The authors characterized in depth one of the prefoldin subunits, Pfdn5 and convincingly demonstrated that this protein functions in regulation of microtubule organization, likely due to its regulation of proper folding of tubulin monomers. They demonstrate convincingly using both immunohistochemistry in larval motor neurons and microtubule binding assays that Pfdn5 is a bona fide microtubule associated protein contributing to the stability of the axonal microtubule cytoskeleton, which is significantly disrupted in the mutants.

Similar phenotypes were observed in larvae expressing the Frontotemporal dementia with Parkinsonism on chromosome 17-associated mutations of the human Tau gene V377M and R406W. On the strength of the phenotypic evidence and the enhancement of the TauV377M-induced eye degeneration they demonstrate that loss of Pfdn5 exaggerates the synaptic deficits upon expression of the Tau mutants. Conversely, overexpression of Pfdn5 or Pfdn6 ameliorates the synaptic phenotypes in the larvae, the vacuolization phenotypes in the adult, even memory defects upon TauV377M expression.

Strengths:

The phenotypic analyses of the mutant and its interactions with TauV377M at the cell biological, histological, and behavioral levels are precise, extensive, and convincing and achieve the aims of characterization of a novel function of Pfdn5.

Regarding this memory defect upon V377M tau expression. Kosmidis et al (2010) pmid: 20071510, demonstrated that pan-neuronal expression of TauV377M disrupts the organization of the mushroom bodies, the seat of long-term memory in odor/shock and odor/reward conditioning. If the novel memory assay the authors use depends on the adult brain structures, then the memory deficit can be explained in this manner.

If the mushroom bodies are defective upon TauV377M expression does overexpression of Pfdn5 or 6 reverse this deficit? This would argue strongly in favor of the microtubule stabilization explanation.

The discovery that Pfdn5 (and 6 most likely) affect tauV377M toxicity is indeed a novel and important discovery for the Tauopathies field. It is important to determine whether this interaction affects only the FTDP-17-linked mutations, or also WT Tau isoforms, which are linked to the rest of the Tauopathies. Also, insights on the mode(s) that Pfdn5/6 affect Tau toxicity, such as some of the suggestions above are aiming at, will likely be helpful towards therapeutic interventions.

Weaknesses:

What is unclear however is how Pfdn5 loss or even overexpression affects the pathological Tau phenotypes.

Does Pfdn5 (or 6) interact directly with TauV377M? Colocalization within tissues is a start, but immunoprecipitations would provide additional independent evidence that this is so.

Does Pfdn5 loss exacerbate TauV377M phenotypes because it destabilizes microtubules, which are already at least partially destabilized by Tau expression?

Rescue of the phenotypes by overexpression of Pfdn5 agrees with this notion.

However, Cowan et al (2010) pmid: 20617325 demonstrated that wild-type Tau accumulation in larval motor neurons indeed destabilizes microtubules in a Tau phosphorylation-dependent manner.

So, is TauV377M hyperphosphorylated in the larvae?? What happens to TauV377M phosphorylation when Pfdn5 is missing and presumably more Tau is soluble and subject to hyperphosphorylation as predicted by the above?

Expression of WT human Tau (which is associated with most common Tauopathies other than FTDP-17) as Cowan et al suggest has significant effects on microtubule stability, but such Tau-expressing larvae are largely viable. Will one mutant copy of the Pfdn5 knockout enhance the phenotype of these larvae?? Will it result in lethality? Such data will serve to generalize the effects of Pfdn5 beyond the two FDTP-17 mutations utilized.

Does the loss of Pfdn5 affect TauV377M (and WTTau) levels?? Could the loss of Pfdn5 simply result in increased Tau levels? And conversely, does overexpression of Pfdn5 or 6 reduce Tau levels?? This would explain the enhancement and suppression of TauV377M (and possibly WT Tau) phenotypes. It is an easily addressed, trivial explanation at the observational level, which if true begs for a distinct mechanistic approach.

Finally, the authors argue that TauV377M forms aggregates in the larval brain based on large puncta observed especially upon loss of Pfdn5. This may be so, but protocols are available to validate this molecularly the presence of insoluble Tau aggregates (for example, pmid: 36868851) or soluble Tau oligomers as these apparently differentially affect Tau toxicity. Does Pfdn5 loss exaggerate the toxic oligomers and overexpression promotes the more benign large aggregates??

Comments on revisions:

In the revised manuscript Βisht et al have provided extensive new experimental evidence in support of previously more tenuous claims. These fully satisfy my comments and suggestions, and in my view, have significantly strengthened the manuscript with compelling new evidence.

---

## [Referee Report · Reviewer #2 (Public review)]

Bisht et al detail a novel interaction between the chaperone, Prefoldin 5, microtubules, and tau-mediated neurodegeneration, with potential relevance for Alzheimer's disease and other tauopathies. Using Drosophila, the study shows that Pfdn5 is a microtubule-associated protein, which regulates tubulin monomer levels and can stabilize microtubule filaments in the axons of peripheral nerves. The work further suggests that Pfdn5/6 may antagonize Tau aggregation and neurotoxicity. While the overall findings may be of interest to those investigating the axonal and synaptic cytoskeleton, the detailed mechanisms for the observed phenotypes remain unresolved and the translational relevance for tauopathy pathogenesis is yet to be established. Further, a number of key controls and important experiments are missing that are needed to fully interpret the findings.

The strength of this study is the data showing that Pfdn5 localizes to axonal microtubules and the loss-of-function phenotypic analysis revealing disrupted synaptic bouton morphology. The major weakness relates to the experiments and claims of interactions with Tau-mediated neurodegeneration. In particular, it is unclear whether knockdown of Pfdn5 may cause eye phenotypes independent of Tau. Further, the GMR>tau phenotype appears to have been incorrectly utilized to examine age-dependent, neurodegeneration.

This manuscript argues that its findings may be relevant to thinking about mechanisms and therapies applicable to tauopathies; however, this is premature given that many questions remain about the interactions from Drosophila, the detailed mechanisms remain unresolved, and absent evidence that tau and Pfdn may similarly interact in the mammalian neuronal context. Therefore, this work would be strongly enhanced by experiments in human or murine neuronal culture or supportive evidence from analyses of human data.

Comments on revisions:

The revision adequately addresses most of the previously raised concerns, resulting in a significantly improved manuscript.

---

## [Author Response]

The following is the authors’ response to the original reviews.

**Reviewer #1 (Public Review):**
Summary:In this manuscript, Bisht et al address the hypothesis that protein folding chaperones may be implicated in aggregopathies and in particular Tau aggregation, as a means to identify novel therapeutic routes for these largely neurodegenerative conditions.The authors conducted a genetic screen in the Drosophila eye, which facilitates the identification of mutations that either enhance or suppress a visible disturbance in the nearly crystalline organization of the compound eye. They screened by RNA interference all 64 known Drosophila chaperones and revealed that mutations in 20 of them exaggerate the Tau-dependent phenotype, while 15 ameliorated it. The enhancer of the degeneration group included 2 subunits of the typically heterohexameric prefoldin complex and other co-translational chaperones.The authors characterized in depth one of the prefoldin subunits, Pfdn5, and convincingly demonstrated that this protein functions in the regulation of microtubule organization, likely due to its regulation of proper folding of tubulin monomers. They demonstrate convincingly using both immunohistochemistry in larval motor neurons and microtubule binding assays that Pfdn5 is a bona fide microtubule-associated protein contributing to the stability of the axonal microtubule cytoskeleton, which is significantly disrupted in the mutants.Similar phenotypes were observed in larvae expressing Frontotemporal dementia with Parkinsonism on chromosome 17-associated mutations of the human Tau gene V377M and R406W. On the strength of the phenotypic evidence and the enhancement of the TauV377Minduced eye degeneration, they demonstrate that loss of Pfdn5 exaggerates the synaptic deficits upon expression of the Tau mutants. Conversely, the overexpression of Pfdn5 or Pfdn6 ameliorates the synaptic phenotypes in the larvae, the vacuolization phenotypes in the adult, and even memory defects upon TauV377M expression.StrengthsThe phenotypic analyses of the mutant and its interactions with TauV377M at the cell biological, histological, and behavioral levels are precise, extensive, and convincing and achieve the aims of characterization of a novel function of Pfdn5.Regarding this memory defect upon V377M tau expression. Kosmidis et al (2010), PMID: 20071510, demonstrated that pan-neuronal expression of Tau^V377M^ disrupts the organization of the mushroom bodies, the seat of long-term memory in odor/shock and odor/reward conditioning. If the novel memory assay the authors use depends on the adult brain structures, then the memory deficit can be explained in this manner.(1) If the mushroom bodies are defective upon Tau^V377M^. expression, does overexpression of Pfdn5 or 6 reverse this deficit? This would argue strongly in favor of the microtubule stabilization explanation.

We thank the reviewer for this insightful comment. Consistent with Kosmidis et al. (2010), we confirm that expression of hTau^V377M^ disrupts the architecture of mushroom bodies. In addition, we find, as suggested by the reviewer, that coexpression of either Pfdn5 or Pfdn6 with hTau^V377M^ significantly restores the organization of the mushroom bodies. These new findings strongly support the hypothesis that Pfdn5 or Pfdn6 mitigate hTau^V377M^ -induced memory deficits by preserving the structure of the mushroom body, likely through stabilizing the microtubule network. This data has now been included in the revised manuscript (Figure 7H-O).

(2) The discovery that Pfdn5 (and 6 most likely) affects tauV377M toxicity is indeed a novel and important discovery for the Tauopathies field. It is important to determine whether this interaction affects only the FTDP-17-linked mutations or also WT Tau isoforms, which are linked to the rest of the Tauopathies. Also, insights on the mode(s) that Pfdn5/6 affect Tau toxicity, such as some of the suggestions above, are aiming at will likely be helpful towards therapeutic interventions.

We agree that determining whether prefoldin modulates the toxicity of both mutant and wildtype Tau is critical for understanding its broader relevance to Tauopathies. We have now performed additional experiments required to address this issue. These new data show that loss of Pfdn5 also exacerbates toxicity associated with wildype Tau (hTau^WT^), in a manner similar to that observed with hTau^V337M^ or hTau^R406W^. Specifically, overexpression of hTau^WT^ in a Pfdn5 mutant background leads to Tau aggregate formation (Figure S7G-I), and coexpression of Pfdn5 with hTau^WT^ reduces the associated synaptic defects (Figure S11F-L). These findings underscore a general role for Pfdn5 in modulating diverse Tauopathy-associated phenotypes and suggest that it could be a broadly relevant therapeutic target.

Weakness(3) What is unclear, however, is how Pfdn5 loss or even overexpression affects the pathological Tau phenotypes. Does Pfdn5 (or 6) interact directly with TauV377M? Colocalization within tissues is a start, but immunoprecipitations would provide additional independent evidence that this is so.

We appreciate this important suggestion. To investigate a potential direct interaction between Pfdn5 and Tau^V377M^, we performed co-immunoprecipitation experiments using lysates from adult fly brain expressing hTau^V337M^. Under the conditions tested, we did not detect a direct physical interaction. While this does not support a direct interaction, it does not strongly refute it either. We note that Pfdn5 and Tau are colocalized within axons (Figure S13J-K). At this stage, we are unable to resolve the issue of direct vs indirect association. If indirect, then Tau and Pfdn5 act within the same subcellular compartments (axon); if direct, then either only a small fraction of the total cellular proteins is in the Tau-Pfdn5 complex and therefore difficult to detect in bulk protein westerns, or the interactions are dynamic or occur in conditions that we have not been able to mimic in vitro.

(4) Does Pfdn5 loss exacerbate Tau^V377M^ phenotypes because it destabilizes microtubules, which are already at least partially destabilized by Tau expression? Rescue of the phenotypes by overexpression of Pfdn5 agrees with this notion.However, Cowan et al (2010) pmid: 20617325 demonstrated that wildtype Tau accumulation in larval motor neurons indeed destabilizes microtubules in a Tau phosphorylation-dependent manner. So, is Tau^V377M^ hyperphosphorylated in the larvae?? What happens to Tau^V377M^ phosphorylation when Pfdn5 is missing and presumably more Tau is soluble and subject to hyperphosphorylation as predicted by the above?

We completely agree that it is important to link Tau-induced phenotypes with the microtubule destabilization and phosphorylation state of Tau. We performed immunostaining using futsch antibody to check the microtubule organization at the NMJ and observed a severe reduction in futsch intensity when Tau^V337M^ was expressed in the Pfdn5 mutant (ElavGal4>Tau^V337M^; DPfdn5^15/40^), suggesting that Pfdn5 absence exacerbates the hTau^V337M^ defects due to more microtubule destabilization (Figure S6F-J).

We have performed additional experiments to examine the phosphorylation state of hTau in Drosophila larval axons. Immunocytochemistry indicated that only a subset of hTau aggregates in Pfdn5 mutants (Elav-Gal4>Tau^V337M^; DPfdn5^15/40^) are recognized by phospho-hTau antibodies. For instance, the AT8 antibody (targeting pSer202/pThr205) (Goedert et al., 1995) labelled only a subset of aggregates identified by the total hTau antibody (D5D8N) (Figure S9AE). Moreover, feeding these larvae (Elav-Gal4>Tau^V337M</sup; DPfdn515/40) with LiCl, which blocks GSK3b, still showed robust Tau aggregation (Figure S9F-J).^

These results imply that: (a) soluble phospho-hTau levels in Pfdn5 mutants are low and not reliably detected with a single phospholylation-specific antibody; (b) Loss of Pfdn5 results in Tau aggregation in a hyperphosphorylation-independent manner similar to what has been reported earlier (LI et al. 2022); and (c) the destabilization of microtubules in Elav-Gal4>Tau^V337M^; DPfdn5^15/40^ results in Tau dissociation and aggregate formation. These data and conclusions have been incorporated into the revised manuscript.

(5) Expression of WT human Tau (which is associated with most common Tauopathies other than FTDP-17) as Cowan et al suggest has significant effects on microtubule stability, but such Tauexpressing larvae are largely viable. Will one mutant copy of the Pfdn5 knockout enhance the phenotype of these larvae?? Will it result in lethality? Such data will serve to generalize the effects of Pfdn5 beyond the two FDTP-17 mutations utilized.

We have now examined whether heterozygous loss of Pfdn5 (∆Pfdn5/+) enhances the effect of Tau expression. While each genotype (hTau^V337M^, hTau^WT^ or ∆Pfdn5/+) alone is viable, Elav-Gal4 driven expression of hTau^V337M^ or hTau^WT^ in Pfdn5 heterozygous background does not cause lethality.

(6) Does the loss of Pfdn5 affect TauV377M (and WTTau) levels?? Could the loss of Pfdn5 simply result in increased Tau levels? And conversely, does overexpression of Pfdn5 or 6 reduce Tau levels?? This would explain the enhancement and suppression of Tau^V377M^ (and possibly WT Tau) phenotypes. It is an easily addressed, trivial explanation at the observational level, which, if true, begs for a distinct mechanistic approach.

To test whether Pfdn5 modulates Tau phenotypes by altering Tau protein levels, we performed western blot analysis under Pfdn5 or Pfdn6 overexpression conditions and observed no change in hTau^V337M^ levels (Figure 6O). However, in the absence of Pfdn5, both hTau^V337M^ and hTau^WT^ form large, insoluble aggregates that are not detected in soluble lysates by standard western blotting but are visualized by immunocytochemistry (Figure S7G-I). Thus, the apparent reduction in Tau levels on western blots reflects a solubility shift, not an actual decrease in Tau expression. These findings argue against a simple model in which Pfdn5 regulates Tau abundance and instead support a mechanism in which Pfdn5 loss leads to change in Tau conformation, leading to its sequesteration away for already destabilized microtubules.

(7) Finally, the authors argue that Tau^V377M^ forms aggregates in the larval brain based on large puncta observed especially upon loss of Pfdn5. This may be so, but protocols are available to validate this molecularly the presence of insoluble Tau aggregates (for example, pmid: 36868851) or soluble Tau oligomers, as these apparently differentially affect Tau toxicity. Does Pfdn5 loss exaggerate the toxic oligomers, and overexpression promote the more benign large aggregates??

We have performed additional experiments to analyze the nature of these aggregates using 1,6-HD. The 1,6-hexanediol can dissolve the Tau aggregate seeds formed by Tau droplets, but cannot dissolve the stable Tau aggregates (WEGMANN et al. 2018). We observed that 5% 1,6hexanediol failed to dissolve these Tau aggregates (Figure S8), demonstrating the formation of stable filamentous flame-shaped NFT-like aggregates in the absence of Pfdn5 (Figure 5D and Figure S9).

**Reviewer #2 (Public review):**
Bisht et al detail a novel interaction between the chaperone, Prefoldin 5, microtubules, and taumediated neurodegeneration, with potential relevance for Alzheimer's disease and other tauopathies. Using Drosophila, the study shows that Pfdn5 is a microtubule-associated protein, which regulates tubulin monomer levels and can stabilize microtubule filaments in the axons of peripheral nerves. The work further suggests that Pfdn5/6 may antagonize Tau aggregation and neurotoxicity. While the overall findings may be of interest to those investigating the axonal and synaptic cytoskeleton, the detailed mechanisms for the observed phenotypes remain unresolved and the translational relevance for tauopathy pathogenesis is yet to be established. Further, a number of key controls and important experiments are missing that are needed to fully interpret the findings.The strength of this study is the data showing that Pfdn5 localizes to axonal microtubules and the loss-of-function phenotypic analysis revealing disrupted synaptic bouton morphology. The major weakness relates to the experiments and claims of interactions with Tau-mediated neurodegeneration.In particular, it is unclear whether knockdown of Pfdn5 may cause eye phenotypes independent of Tau.

Our new experiments confirm that knockdown of Pfdn5 alone does not cause eye phenotypes.

Further, the GMR>tau phenotype appears to have been incorrectly utilized to examine agedependent, neurodegeneration.

In response, we have modulated and explained our conclusions in this regard as described later in our “rebuttal.”

This manuscript argues that its findings may be relevant to thinking about mechanisms and therapies applicable to tauopathies; however, this is premature given that many questions remain about the interactions from Drosophila, the detailed mechanisms remain unresolved, and absent evidence that Tau and Pfdn may similarly interact in the mammalian neuronal context. Therefore, this work would be strongly enhanced by experiments in human or murine neuronal culture or supportive evidence from analyses of human data.

The reviewer is correct that the impact would be greater if Pfdn5-Tau interactions were also examined in human tissue. While we have not attempted these experiments ourselves, we hope that our observations will stimulate others to test the conservation of phenomena we describe. There are, however, several lines of circumstantial evidence from human Alzheimer’s disease datasets that implicate PFDN5 in disease pathology. For example, recent compilations and analyses of proteomic data show reductions of CCT components, TBCE, as well as Prefoldin subunits, including PFDN5, in AD tissue (HSIEH et al. 2019; TAO et al. 2020; JI et al. 2022; ASKENAZI et al. 2023; LEITNER et al. 2024; SUN et al. 2024). Furthermore, whole blood mRNA expression data from Alzheimer's patients revealed downregulation of PFDN5 transcript (JI et al. 2022). Together, these findings from human data are consistent with the roles of PFDN5 in suppressing diverse neurodegenerative processes. We have incorporated these points into the discussion section of the revised manuscript.

**Reviewer #1 (Recommendations for the authors):**
See public review for experimental recommendations focusing on the Tau Pfdn interactions. I would refrain from using the word aggregates, I would call them puncta, unless there is molecular or visual (ie AFM) evidence that they are indeed insoluble aggregates. Finally, although including the full genotypes written out below the axis in the bar graphs is appreciated, it nevertheless makes them difficult to read due to crowding in most cases and somewhat distracting from the figure.In my opinion, a more reader-friendly manner of reporting the phenotypes will be highly helpful. For example, listing each component of the genotype on the left of each bar graph and adding a cross or a filled circle under the bar to inform of the full genotype of the animals used.

As described in the response to the previous comment, we now have strong direct evidences to support our view that the observed puncta are stable Tau aggregates. Thus, we feel justified to use the term Tau-aggregates in preference to Tau puncta.

We have tried to write the genotypes to make them more reader-friendly.

**Reviewer #2 (Recommendations for the authors):**
(1) Lines 119-121: 35 modifiers from 64 seem like an unusually high hit rate. Are these individual genes or lines? Were all modifiers supported by at least 2 independent RNAi strains targeting non-overlapping sequences? A supplemental table should be included detailing all genes and specific strains tested, with corresponding results.

We agree with the reviewer that 35 modifiers from 64 genes may be too high. However, since the genes knocked down in the study are chaperones, crucial for maintaining proteostasis, we may have got unusually high hits. The information related to individual genes and lines is provided in Supplemental Table 1. We have now included an additional Supplemental Table 3, which lists the genes and the RNAi lines used in Figure 1, detailing the sequence target information. The table also specifies the number of independent RNAi strains used and the corresponding results.

(2) Figure 1: The authors quantify the areas of ommatidial fusion and necrosis as degeneration, but it is difficult to appreciate the aberrations in the photos provided. Was any consideration given to also quantifying eye size?

We have processed the images to enhance their contrast and make the aberrations clearer. The percentage of degenerated eye area (Figure 1M) was normalized with total eye area. The method for quantifying degenerated area has been explained in the materials and methods section.

(3) Figure 1: (a) Only enhancers of rough eyes are shown but no controls are included to evaluate whether knockdown of these genes causes eye toxicity in the absence of Tau. These are important missing controls. All putative Tau enhancers, including Pdn5/6, need to be tested with GMR-GAL4 independently of Tau to determine whether they cause a rough eye. In a previous publication from some of the same investigators (Raut et al 2017), knockdown of Pfdn using eyGAL4 was shown to induce severe eye morphology defects - this raises questions about the results shown here.

We agree that assessing the effects of HSP knockdown independent of Tau is essential to confirm modifier specificity. We have now performed these knockdowns, and the data are reported in Supplemental Table 1. For RNAi lines represented in Figure 1, which enhanced Tau-induced degeneration/eye developmental defect, except for one of the RNAi lines against Pfdn6 (GD34204), no detectable eye defects were observed when knocked down with GMR-Gal4 at 25°C, suggesting that enhancement is specific to the Tau background.

Use of a more eye-specific GMR-Gal4 driver at 25°C versus broader expressing ey-Gal4 at 29°C in prior work (Raut et al. 2017) likely reflects the differences in the eye morphological defects.

(b) Besides RNAi, do the classical Pdn5 deletion alleles included in this work also enhance the tau rough eye when heterozygous? Please also consider moving the Pfdn5/6 overexpression studies to evaluate possible suppression of the Tau rough eye to Figure 1, as it would enhance the interpretation of these data (but see also below).

GMR-Gal4 driven expression of hTau^V337M^ or hTau^WT^ in Pfdn5 heterozygous background does not enhance rough eye phenotype.

(4) For genes of special interest, such as Pdn5, and other genes mentioned in the results, the main figure, or discussion, it is also important to perform quantitative PCR to confirm that the RNAi lines used actually knock down mRNA expression and by how much. These studies will establish specificity.

We agree that confirming RNAi efficiency via quantitative PCR (qPCR) is essential for validating the knockdown efficiency. We have now included qPCR data, especially for key modifiers, confirming effective knockdown (Figure S2).

(5) Lines 235-238: how do you conclude whether the tau phenotype is "enhanced" when Pfdn5 causes a similar phenotype on its own? Could the combination simply be additive? Did overexpression of Pdn5 suppress the UAS-hTau NMJ bouton phenotype (see below)?

Although Pfdn5 mutants and hTau expression individually increase satellite boutons, their combination leads to a significantly more severe and additional phenotype, such as significantly decreased bouton size and increased bouton number, indicating an enhancing rather than purely additive interaction (Figure 4 and Figure S6C). Moreover, we now show that overexpression of Pfdn5 significantly suppressed the hTau^V337M^-induced NMJ phenotypes. This new data has been incorporated as Figure S11F-L in the revised manuscript.

Alternatively, did the authors consider reducing fly tau in the Pdn5 mutant background?

In new additional experiments, we observe that double mutants for Drosophila Tau (dTau) and Pfdn5 also exhibit severe NMJ defects, suggesting genetic interactions between dTau and Pfdn5. This data is shown below for the reviewer.

**Author response image 1. sa3fig1:** A double mutant combination of dTau and Pfdn5 aggravates the synaptic defects at the Drosophila NMJ. (A-D') Confocal images of NMJ synapses at muscle 4 of A2 hemisegment showing synaptic morphology in (A-A') control, (B-B') ΔPfdn5^15/40^, (C-C') dTauKO/dTauKO (Drosophila Tau mutant), (D-D') dTauKO/dTauKO; ∆Pfdn5^15/40^ double immunolabeled for HRP (green), and CSP (magenta). The scale bar in D for (A-D') represents 10 µm.

(6) It may be important to further extend the investigation to the actin cytoskeleton. It is noted that Pfdn5 also stabilizes actin. Importantly, tau-mediated neurodegeneration in Drosophila also disrupts the actin cytoskeleton, and many other regulators of actin modify tau phenotypes.

We appreciate the suggestion to examine the actin cytoskeleton. While prior studies indicate that Pfdn5 might regulate the actin cytoskeleton and that Tau^V377M^ hyperstabilizes the actin cytoskeleton, we did not observe altered actin levels in Pfdn5 mutants (Figure 2G). However, actin dynamics may represent an additional mechanism through which Pfdn5 might temporally influence Tauopathy. Future work will address potential actin-related mechanisms in Tauopathy.

(7) Figure 2: in the provided images, it is difficult to appreciate the futsch loops. Please include an image with increased magnification. It appears that fly strains harboring a genomic rescue BAC construct are available for Pfdn-this would be a complementary reagent to test besides Pfdn overexpression.

We have updated Figure 2 to include high magnification NMJ images as insets, clearly showing the Futsch loops. While we have not yet tested a genomic rescue BAC construct for Pfdn5, we plan to use the fly line harboring this construct in future work.

(8) Figure 3: Some of the data is not adequately explained. The use of Ran as a loading control seems rather unusual. What is the justification? Pfdn appears to only partially co-localize with a-tubulin in the axon; can the authors discuss or explain this? Further, in Pfdn5 mutants, there appears to be a loss of a-tubulin staining (3b'); this should also be discussed.

We appreciate the reviewer's concern regarding the choice of loading control for our Western blot analysis. Importantly, since Tubulin levels and related pathways were the focus of our analysis, traditional loading controls such as α- or β-tubulin or actin were deemed unsuitable due to potential co-regulation. Ran, a nuclear GTPase involved in nucleocytoplasmic transport, is not known to be transcriptionally or post-translationally regulated by Tubulin-associated signaling pathways. To ensure its reliability as a loading control, we confirmed by densitometric analysis that Ran expression showed minimal variability across all samples. Hence, we used Ran for accurate normalization in the Western blot data represented in this manuscript. We have also used GAPDH as a loading control and found no difference with respect to Ran as a loading control across samples.

We appreciate the reviewer's comment regarding the interpretation of our Pearson's correlation coefficient (PCC) results. While the mean colocalization value of 0.6 represents a moderate positive correlation (MUKAKA 2012), which may not reach the conventional threshold for "high positive" colocalization (usually considered 0.7-0.9), it nonetheless indicates substantial spatial overlap between the proteins of interest. Importantly, colocalization analysis provides supportive but indirect evidence for molecular proximity. To further validate the interaction, we performed a microtubule binding assay, which directly demonstrates the binding of Pfdn5 to stabilized microtubules.

In accordance with the western blot analysis shown in Figure 2G-I, the levels of Tubulin are reduced in the Pfdn5 mutants (Figure 3B''). We have incorporated and discussed this in the revised manuscript.

(9) Figure 4: Overexpression of Pfdn appears to rescue the supernumerary satellite bouton numbers induced by human Tau; however, interpretation of this experiment is somewhat complicated as it is performed in Pfdn mutant genetic background. Can overexpression of Pfdn on its own rescue the Tau bouton defect in an otherwise wildtype background?

We have now coexpressed Pfdn5 and hTau^V337M^ in an otherwise wild-type background. As shown in Figure S11F-L, Pfdn5 overexpression suppresses Tau-induced bouton defects. We have incorporated the data in the Results section to support the role of Pfdn5 as a modifier of Tau toxicity.

(10) Lines 256-263 / Figure 5: (a) What exactly are these tau-positive structures (punctae) being stained in larval brains in Fig 5C-E? Most prior work on tau aggregation using Drosophila models has been done in the adult brain, and human wildtype or mutant Tau is not known to form significant numbers of aggregates in neurons (although aggregates have been described following glia tau expression).Therefore, the results need to be further clarified. Besides the provided schematic, a zoomed-out image showing the whole larval brain is needed here for orientation. Have these aggregates been previously characterized in the literature?

We agree with the reviewer that the expression of the wildtype or mutant form of human Tau in Drosophila is not known to form aggregates in the larval brain, in contrast to the adult brain (JACKSON et al. 2002; OKENVE-RAMOS et al. 2024). Consistent with previous reports, we also observed that Tau expression on its own does not form aggregates in the Drosophila larval brain.

However, in the absence of Pfdn5, microtubule disruption is severe, leading to reduced Taumicrotubule binding and formation of globular/round or flame-shaped tangles like aggregates in the larval brain. Previous studies have reported that 1,6-hexanediol can dissolve the Tau aggregate seeds formed by Tau droplets, but cannot dissolve the stable Tau aggregates (WEGMANN et al. 2018). We observed that 5% 1,6-Hexanediol failed to dissolve these Tau puncta, demonstrating the formation of stable aggregates in the absence of Pfdn5. Additionally, we now performed a Tau solubility assay and show that in the absence of Pfdn5, a significant amount of Tau goes in the pellet fraction, which could not be detected by phospho-specific AT8 Tau antibody (targeting pSer202/pThr205) but was detected by total hTau antibody (D5D8N) on the western blots (Figure S8). These data further reinforce our conclusion that Pfdn5 prevents the transition of hTau from soluble and/or microtubule-associated state to an aggregated, insoluble, and pathogenic state. These new data have been incorporated into the revised manuscript.

(b) Can additional markers (nuclei, cell membrane, etc.) be used to highlight whether the taupositive structures are present in the cell body or at synapses?

We performed the co-staining of Tau and Elav to assess the aggregated Tau localization. We found that in the presence of Pfdn5, Tau is predominantly cytoplasmic and localised to the cell body and axons. In the absence of Pfdn5, Tau forms aggregates but is still localized to the cell body or axons. However, some of the aggregates are very large, and the subcellular localization could not be determined (Figure S8M-N'). These might represent brain regions of possible nuclear breakdown and cell death (JACKSON et al. 2002).

(c) It would also be helpful to perform western blots from larval (and adult) brains examining tau protein levels, phospho-tau species, possible higher-molecular weight oligomeric forms, and insoluble vs. soluble species. These studies would be especially important to help interpret the potential mechanisms of observed interactions.

Western blot analysis revealed that overexpression of Pfdn5 does not alter total Tau levels (Figure 6O). In Pfdn5 mutants, however, hTau^V337M^ levels were reduced in the supernatant fraction and increased in the pellet fraction, indicating a shift from soluble monomeric Tau to aggregated Tau.

(d) Does overexpression of Pdn5 (UAS-Pdn5) suppress the formation of tau aggregates? I would therefore recommend that additional experiments be performed looking at adult flies (perhaps in Pfdn5 heterozygotes or using RNAi due to the larval lethality of Pdn5 null animals).

Overexpression of Pfdn5 significantly reduced Tau-aggregates (Elav-Gal4/UASTau^V337M^; UAS-Pfdn5; DPfdn5^15/40^) observed in Pfdn5 mutants (Figure 5E). Coexpression of Pfdn5 and hTau^V337M^ suppresses the Tau aggregates/puncta in 30-day adult brain. Since heterozygous DPfdn^15^/+ did not show a reduction in Pfdn5 levels, we did not test the suppression of Tau aggregates in DPfdn^15^/+; Elav>UAS-Pfdn5, UAS-Tau^V337M^.

(11) Figure 6, panels A-N: The GMR>Tau rough eye is not a "neurodegenerative" but rather a predominantly developmental phenotype. It results from aberrant retinal developmental patterning and the subsequent secretion/formation of the overlying eye cuticle (lenslets). I am confused by the data shown suggesting a "shrinking eye size" and increasing roughened surface over time (a GMR>tau eye similar to that shown in panel B cannot change to appear like the one in panel H with aging). The rough eye can be quite variable among a population of animals, but it is usually fixed at the time the adult fly ecloses from the pupal case, and quite stable over time in an individual animal. Therefore, any suppression of the Tau rough eye seen at 30 days should be appreciable as soon as the animals eclose. These results need to be clarified. If indeed there is robust suppression of Tau rough eye, it may be more intuitive and clearer to include these data with Figure 1, when first showing the loss-of-function enhancement of the Tau rough eye. Also, why is Pfdn6 included in these experiments but not in the studies shown in Figures 2-5?

We thank the reviewer for their careful and knowledgeable assessment of the GMR>Tau rough eye model. We appreciate the clarification that the rough eye phenotype could be “developmental” rather than neurodegenerative.” Our initial observations regarding "shrinking eye size" and "increased surface roughness" clearly show age-related progression of structural change. Such progression has been observed and reported by others (IIJIMA-ANDO et al. 2012; PASSARELLA AND GOEDERT 2018). We observed an age-dependent increase in the number of fused ommatidia in GMR-Gal4 >Tau, which were rescued by Pfdn5 or Pfdn6 expression. We noted that adult-specific induction of hTau^V337M^ adult flies using the Gal80^ts^ and GMR-GeneSwitch (GMR-GS) systems was not sufficient to induce a significant eye phenotype; thus, early expression of Tau in the developing eye imaginal disc appears to be required for the adult progressive phenotype that we observe. We feel that it is inadequate to refer to this adult progressive phenotype as “developmental,” and while admittedly arguable whether this can be termed “degenerative.”

To address neurodegeneration more directly, we focused on 30-day-old adult fly brains and demonstrated that Pfdn5 overexpression suppresses age-dependent Tau-induced neurodegeneration in the central nervous system (Figure 6H-N and Figure S12). This supports our central conclusion regarding the neuroprotective role of Pfdn5 in age-associated Tau pathology. Since we found an enhancement in the Tau-induced synaptic and eye phenotypes by Pfdn6 knockdown, we also generated CRISPR/Cas9-mediated loss-of-function mutants for Pfdn6. However, loss of Pfdn6 resulted in embryonic/early first instar lethality, which precluded its detailed analysis at the larval stages.

(12) Figure 6, panels O-T: the elav>tau image appears to show a different frontal section plane compared to the other panels. It is advisable to show images at a similar level in all panels since vacuolar pathology can vary by region. It is also useful to be able to see the entire brain at a lower power, but the higher power inset view is obscuring these images. I would recommend creating separate panels rather than showing them as insets.

In the revised figure, we now display the low- and high-magnification images as separate, clearly labeled panels instead of using insets. This improves visibility of the brain morphology while providing detailed views of the vacuolar pathology (Figure 6H-L).

(13) Figure 6/7: For the experiments in which Pfdn5/6 is overexpressed and possibly suppresses tau phenotypes (brain vacuoles and memory), it is important to use controls that normalize the number of UAS binding sites, since increased UAS sites may dilute GAL4 and reduced Tau expression levels/toxicity. Therefore, it would be advisable to compare with Elav>Tau flies that also include a chromosome with an empty UAS site or other transgenes, such as UAS-GFP or UAS-lacZ.

We thank the reviewer for the suggestion. Now we have incorporated proper controls in the brain vacuolization, the mushroom body, and ommatidial fusion rescue experiments. Also, we have independently verified whether Gal4 dilution has any effect on the Tau phenotypes (Figure 6H-L, Figure 7, and Figure S11A-B).

(14) Lines 311-312: the authors say vacuolization occurs in human neurodegenerative disease, which is not really true to my knowledge and definitely not stated in the citation they use. Please re-phrase.

Now we have made the appropriate changes in the revised manuscript.

(15) Figure 7: The authors claim that Pfdn5/6 expression does not impact memory behavior, but there in fact appears to be a decrease in preference index (panel D vs panel B). Does this result complicate the interpretation of the potential interaction with Tau (panel F). Are data from wildtype control flies available?

In our memory assay, a decrease in performance index (PI) of the trained flies compared to the naïve flies indicates memory formation (normal memory in control flies, Figure 7B). In contrast, a lack of significant difference in PI indicates a memory defect (Figure 7C: hTau^V337M^ overexpressed flies). "Decrease in preference index (panel D vs panel B)" is not a sign of memory defect; it may be interpreted as a better memory instead. Hence, neuronal overexpression of Pfdn5 (Figure 7D) or Pfdn6 (Figure 7E) in wildtype neurons does not cause memory deficits. In addition, coexpression of Pfdn5/6 and hTau^V337M^ successfully rescues the Tau-induced memory defect (significant drop in PI compared to the PI of naïve flies in Figure 7F-G). Moreover, almost complete rescue of the Tau-induced mushroom body defect on Pfdn5 or Pfdn6 expression further establishes potential interaction between Pfdn5/6 and Tau. This data has been incorporated into the revised manuscript.

The memory assay itself with extensive data on wildtype flies and various other genotype will shortly be submitted for publication in another manuscript (Majumder et al, manuscript under preparation); However, we can confirm for the reviewer that wildtype flies, trained and assayed by the protocol described, show a significant decrease in performance index compared to the naïve flies, indicative of strong learning and memory performance, very similar to the control genotype data shown in Figure 7B.

Additional minor considerations(16) Lines 50-52: there are many therapeutic interventions for treating tauopathies, but not curative or particularly effective ones.

Now we have made the appropriate changes in the revised manuscript.

(17) Lines 87-106 seem like a duplication of the abstract. Consider deleting or condensing.

We have made the appropriate changes in the revised manuscript.

(18) Where is pfdn5 expressed? Development v. adult? Neuron v. glia? Conservation?

Prefoldin5 is expressed throughout development but strongly localized to the larval trachea and neuronal axons. Drosophila Pfdn5 shows 35% overall identity with human PFDN5.

(19) Liine 187: is pfdn5 truly "novel"?

The role of Pfdn5 as microtubule-binding and stabilizing is a new finding and has not been predicted or described before. Hence, it is a novel neuronal microtubule-associated protein.

(20) Figure 5, panel F, genotype labels on the x-axis are confusing; consider simplifying to Control, DPfdn, and Rescue.

We have made appropriate changes in the figure for better readability.

(21) Figures 5/8: it might be preferable to use consistent colors for Tau/HRP--Tau is labeled green in Figure 5 and then purple in Figure 8.

We have made these changes where possible.

(22) Lines 311-312: Vacuolar neuropathology is NOT typically observed in human Tauopathy.

We thank the reviewer for pointing this out**.** We have made the appropriate changes in the revised manuscript.

(23) Lines 328-349: The explanation could be made more clear. Naïve flies should not necessarily be called controls. Also, a more detailed explanation of how the preference index is computed would be helpful. Why are some datapoints negative values?

(a) We have rewritten this paragraph to make the description and explanation clearer. The detailed method and formula to calculate the Preference index have been incorporated in the Materials and Methods section.

(b) We have replaced the term Control with Naïve.

(c) Datapoints with negative values appeared in some of the 'Trained' group flies. It indicates that post-CuSO_4_ training, some groups showed repulsion towards the otherwise attractive odor 2,3B. As 2,3B is an attractive odorant, naïve or control flies show attraction towards it compared to air, which is evident from a higher number of flies in the Odor arm (O) compared to that of the Air arm (A) of the Y-maze; thus, the PI [(O-A/O+A)*100] is positive in case of naïve fly groups. Training of the flies led to an association of the attractive odorant with bitter food, leading to a decrease of attraction, and even repulsion towards the odorant in a few instances, resulting in less fly count in the odor arm compared to the air arm. Hence, the PI becomes negative as (O-A) is negative in such instances. Thus, it is not an anomaly but indicates strong learning.

(24) Line 403: misspelling "Pdfn"

We have corrected this.

(25) Lines 423-425: recommend re-phrasing, since tauopathies are human diseases. Mice and other animal models may be susceptible to tau-mediated neuronal dysfunction but not Tauopathy, per see.

We have made the appropriate changes in the revised manuscript.

(26) Lines 468-469: "tau neuropathology" rather than "tau associated neuropathies".

We have made the appropriate changes in the revised manuscript.

References

Askenazi, M., T. Kavanagh, G. Pires, B. Ueberheide, T. Wisniewski et al., 2023 Compilation of reported protein changes in the brain in Alzheimer's disease. Nat Commun 14**:** 4466.

Hsieh, Y. C., C. Guo, H. K. Yalamanchili, M. Abreha, R. Al-Ouran et al., 2019 Tau-Mediated Disruption of the Spliceosome Triggers Cryptic RNA Splicing and Neurodegeneration in Alzheimer's Disease. Cell Rep 29**:** 301-316 e310.

Iijima-Ando, K., M. Sekiya, A. Maruko-Otake, Y. Ohtake, E. Suzuki et al., 2012 Loss of axonal mitochondria promotes tau-mediated neurodegeneration and Alzheimer's disease-related tau phosphorylation via PAR-1. PLoS Genet 8**:** e1002918.

Jackson, G. R., M. Wiedau-Pazos, T. K. Sang, N. Wagle, C. A. Brown et al., 2002 Human wildtype tau interacts with wingless pathway components and produces neurofibrillary pathology in Drosophila. Neuron 34**:** 509-519.

Ji, W., K. An, C. Wang and S. Wang, 2022 Bioinformatics analysis of diagnostic biomarkers for Alzheimer's disease in peripheral blood based on sex differences and support vector machine algorithm. Hereditas 159**:** 38.

Leitner, D., G. Pires, T. Kavanagh, E. Kanshin, M. Askenazi et al., 2024 Similar brain proteomic signatures in Alzheimer's disease and epilepsy. Acta Neuropathol 147**:** 27.

Li, L., Y. Jiang, G. Wu, Y. A. R. Mahaman, D. Ke et al., 2022 Phosphorylation of Truncated Tau Promotes Abnormal Native Tau Pathology and Neurodegeneration. Mol Neurobiol 59**:** 6183-6199.

Mershin, A., E. Pavlopoulos, O. Fitch, B. C. Braden, D. V. Nanopoulos et al., 2004 Learning and memory deficits upon TAU accumulation in Drosophila mushroom body neurons. Learn Mem 11**:** 277-287.

Mukaka, M. M., 2012 Statistics corner: A guide to appropriate use of correlation coefficient in medical research. Malawi Med J 24**:** 69-71.

Okenve-Ramos, P., R. Gosling, M. Chojnowska-Monga, K. Gupta, S. Shields et al., 2024 Neuronal ageing is promoted by the decay of the microtubule cytoskeleton. PLoS Biol 22**:** e3002504.

Passarella, D., and M. Goedert, 2018 Beta-sheet assembly of Tau and neurodegeneration in *Drosophila melanogaster*. Neurobiol Aging 72**:** 98-105.

Sun, Z., J. S. Kwon, Y. Ren, S. Chen, C. K. Walker et al., 2024 Modeling late-onset Alzheimer's disease neuropathology via direct neuronal reprogramming. Science 385**:** adl2992.

Tao, Y., Y. Han, L. Yu, Q. Wang, S. X. Leng et al., 2020 The Predicted Key Molecules, Functions, and Pathways That Bridge Mild Cognitive Impairment (MCI) and Alzheimer's Disease (AD). Front Neurol 11**:** 233.

Wegmann, S., B. Eftekharzadeh, K. Tepper, K. M. Zoltowska, R. E. Bennett et al., 2018 Tau protein liquid-liquid phase separation can initiate tau aggregation. EMBO J 37.